# RELATION-AWARE GRAPH FOUNDATION MODEL FOR FEW-SHOT LEARNING

## ABSTRACT

In recent years, large language models (LLMs) have demonstrated remarkable ability to generalize across diverse natural language processing (NLP) tasks. Inspired by this success, graph foundation models (GFMs) have emerged as a promising direction in graph learning, aiming to achieve cross-dataset generalization through large-scale pre-training. However, unlike language models that rely on explicit token representations, graphs lack a well-defined unit for generalization, making it challenging to design effective pre-training strategies. In this work, we propose REEF, a novel GFM framework that leverages relation tokens as the fundamental units. Analogous to token vocabularies in LLMs, we construct a vocabulary of relation tokens to encode relational information within graphs. To accommodate diverse relations, we introduce two hypernetworks that adaptively generate the parameters of aggregators and classifiers in graph neural networks based on relation tokens. In addition, we design another hypernetwork to construct dataset-specific projectors and incorporate a dataset-level feature bias into the initial node representations, enhancing flexibility across different datasets with the same relation. Furthermore, we adopt graph data augmentation and a mixed-dataset pre-training strategy, allowing REEF to capture relational diversity more effectively. Extensive experiments show that REEF consistently outperforms existing methods on both pre-training and transfer learning tasks, and demonstrates strong generalization in few-shot transfer scenarios, underscoring its potential as a powerful foundation model for graph-based applications.

## 1 INTRODUCTION

In recent years, large language models (LLMs) have gained significant attention for their powerful generalization capabilities (Achiam et al., 2023; Wei et al., 2022; Touvron et al., 2023; Brown et al., 2020). These models, referred to as foundation models, are pre-trained on extensive data, allowing them to learn diverse linguistic patterns and effectively adapt to a wide range of downstream tasks.

Inspired by these advances, the graph learning community is exploring the development of graph foundation models (GFMs), which aim to generalize across diverse datasets through large-scale pre-training and to adapt to few-shot settings with limited supervision. Graphs, as universal data structures, naturally capture entities and their *relations*. By encoding both structural and semantic interactions, these relations form a connective backbone that is intrinsic to graph data. For example, a sentence can be represented as a dependency graph where relations encode grammatical roles (Gao et al., 2021); an image can be modeled as a graph where pixels are connected through spatial adjacency (Wu et al., 2020); and in real-world systems such as citation or social networks (Liu et al., 2022; Myers et al., 2014), relations define how information flows among entities. Unlike language, which explicitly decomposes text into tokens as fundamental units, graph data lacks such clear units of generalization. This absence presents a key challenge for GFMs, making it difficult to design effective pre-training strategies.

Existing attempts often treat either nodes or datasets as fundamental units for pre-training (Huang et al., 2024a; Zhao et al., 2024a; Liu et al., 2024; Xia & Huang, 2024). These approaches either learn distinct node embeddings or rely on mixture-of-experts (MoE) frameworks to handle different datasets. Despite exhibiting some degree of generalization, these methods are limited in capturing universal knowledge, as the information contained in these units exhibits significant heterogeneity.

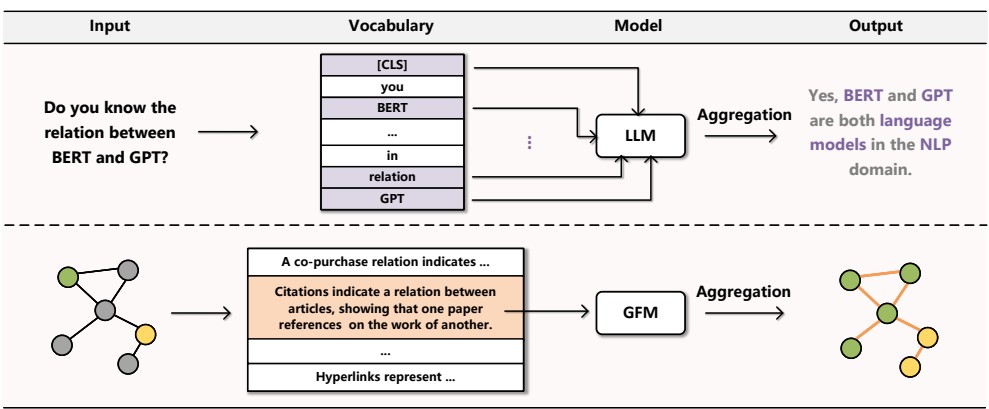

Figure 1: Relation vocabulary inspired by LLMs.

For example, citation networks like Cora and PubMed belong to different domains (computer science vs. biomedicine) and exhibit drastically different features both at the node level and across datasets. In contrast, their *relations* consistently capture citation interactions between nodes across domains. **Therefore, relations offer a more coherent and transferable basis for GFMs, making them better suited as fundamental units for pre-training.**

Our inspiration stems from how LLMs extract linguistic information from tokens, under the premise that a predefined vocabulary encodes each individual token. To comprehend an input sequence and generate appropriate responses, LLMs learn to aggregate information from tokens through weighted attention, yielding a global semantic representation used for output generation. With this idea in mind, we propose extending this paradigm to the graph domain, as depicted in Figure 1. Concretely, we analogously construct a vocabulary of *relation tokens* that encode relational information within graphs, serving as the fundamental units for GFMs. For a given input graph, each type of edge is mapped to a relation token from this vocabulary, capturing the semantics of the relation. A global graph representation is then derived by aggregating messages from the source and target nodes of each relation, weighted by the GFM parameters learned during pretraining. In essence, our approach focuses on modeling relations through flexible, token-specific message aggregation.

Based on this motivation, we propose a novel framework called REEF, which utilizes **RE**lation **E**ncoding to construct a graph **F**oundation model. We conceptualize graph mining tasks as the modeling of diverse relations, where link prediction or classification involves determining whether a relation exists—either between two nodes or between a node and a label. To accommodate various types of relationships, we customize key components of graph neural networks (GNNs), including the *aggregator* and *classifier*. Specifically, we first build a relation vocabulary using language models, encoding textual descriptions of relationships into relational representations. These representations are then used to parameterize two *hypernetworks* (Chauhan et al., 2024; Ha et al., 2016), which adaptively generate relation-specific aggregators and classifiers, enabling flexible message passing and effective task-specific inference. Further, to handle diverse feature distributions across datasets within the same relation, we introduce an additional hypernetwork to construct dataset-specific projectors and incorporate dataset-level bias into the initial node representations. Finally, we adopt pre-training with mixed-dataset training and graph data augmentation strategies, enabling REEF to capture relational diversity more effectively. Extensive experimental results demonstrate that REEF achieves superior performance on both pre-training and transfer learning tasks, and delivers particularly substantial gains in few-shot transfer scenarios. Our main contributions are summarized as follows:

- We propose a novel framework for graph foundation models that leverages relation tokens as the fundamental units, enabling effective pre-training and transfer across datasets.
- We design two hypernetworks to dynamically generate relation-specific aggregators and classifiers based on relational representations. Additionally, we introduce dataset-specific feature projectors and bias terms to enhance representation quality and improve generalization capabilities.
- We conduct extensive experiments demonstrating that REEF achieves state-of-the-art average performance across datasets, with especially strong improvements in few-shot transfer scenarios.

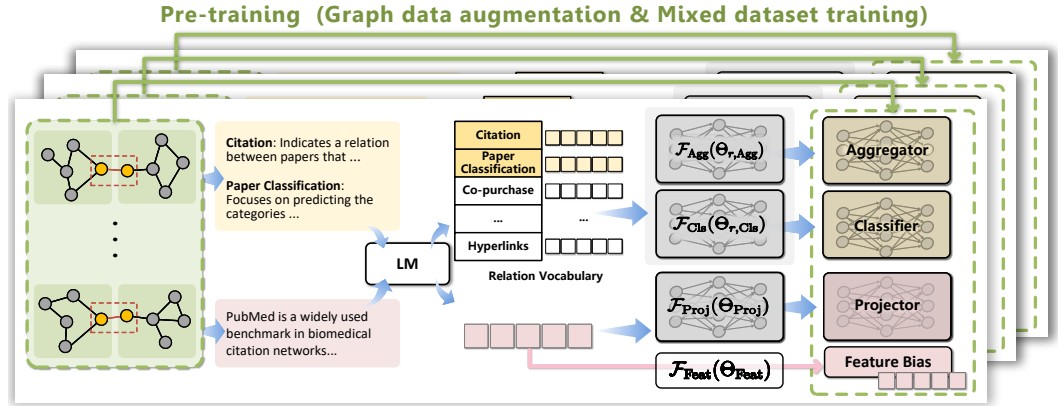

Figure 2: The overall framework of REEF.

## 2 PRELIMINARY

**Graph data.** Each graph dataset is represented as $\mathcal{G} = \{\mathcal{V}_\mathcal{G}, \mathcal{E}_\mathcal{G}, X_\mathcal{G}, \mathcal{R}_\mathcal{G}, t_\mathcal{G}\}$, where $\mathcal{V}_\mathcal{G}$ denotes the set of nodes, $\mathcal{E}_\mathcal{G}$ represents the set of edges, $X_\mathcal{G}$ is the feature matrix corresponding to the nodes, $\mathcal{R}_\mathcal{G}$ is the collection of textual descriptions for different relation types, and $t_\mathcal{G}$ indicates the text description of the dataset. We define $\mathcal{G} \in \mathcal{G}_{\text{source}} \cup \mathcal{G}_{\text{target}}$, where $\mathcal{G}_{\text{source}}$ represents the pretraining graph datasets, $\mathcal{G}_{\text{target}}$ denotes the target graph datasets for transfer learning, and $\mathcal{G}_{\text{source}} \cap \mathcal{G}_{\text{target}} = \emptyset$. Additionally, we define relation vocabulary as $\mathcal{R}_{\text{source}} = \{r | r \in \mathcal{R}_\mathcal{G}, \mathcal{G} \in \mathcal{G}_{\text{source}}\}$, where $r$ represents a relation token, i.e., the textual description of the relation.

**Objective.** The primary objectives of this work are to evaluate two fundamental aspects of Graph Foundation Models (GFMs): (1) *Generalizability:* This aspect focuses on determining whether the model can effectively capture and leverage knowledge acquired from multiple pretraining datasets, enabling it to generalize well across diverse graph structures. (2) *Transferability:* This evaluates the model's ability to transfer the knowledge learned during pretraining to new, unseen datasets, and its effectiveness in adapting to different domains.

**Pre-training Tasks (Relation Token Prediction).** For each node $v_i$, we construct a $k$-hop subgraph $s_i$ and define training triplets in the form $\langle s_i, r_{ij}, s_j \rangle$. Depending on the downstream task, these triplets fall into two categories: (1) Classification tasks: $s_i$ denotes the subgraph centered at the node $v_i$ to be classified, while $s_j$ corresponds to a label node, represented as a subgraph containing only the label node itself. Each label node is defined as the centroid of all training samples associated with that specific label. The relation token $r_{ij}$ denotes the textual description of the classification task within this domain. (2) Link prediction tasks: $s_i$ and $s_j$ are the subgraphs of nodes $v_i$ and $v_j$, respectively, and $r_{ij}$ represents the textual description of the edge type between them. For both tasks, $r_{ij} \in \mathcal{R}_{source}$ represents the potential relationship between the subgraphs $s_i$ and $s_j$. Our pretraining objective is to train a model $\mathcal{F}(\Theta)$ to perform binary classification for different relations, aiming to predict the following probability:

$$P(y = 1 \mid \langle s_i, r_{ij}, s_j \rangle) = \mathcal{F}(\langle s_i, r_{ij}, s_j \rangle; \Theta) , \tag{1}$$

where $P(\cdot \mid \cdot)$ denotes the conditional probability that the relation $r_{ij}$ holds between subgraphs $s_i$ and $s_j$. Thus, the goal of $\mathcal{F}$ is to predict whether this relation exists.

## 3 METHODOLOGY

In this section, we present REEF, whose overall architecture is shown in Figure 2. We first use LMs to generate relational and dataset representations based on their descriptions. These representations are mapped to the parameters of the aggregator and classifier through two hypernetworks, $\mathcal{F}_{\text{Agg}}$ and $\mathcal{F}_{\text{Cls}}$, respectively. The dataset representation is mapped to a feature projector via another hypernetwork $\mathcal{F}_{\text{Proj}}$, and a dataset-specific feature bias is derived to enhance flexibility. Finally, all components are pre-trained with graph data augmentation and a mixed-dataset strategy.

### 3.1 FEATURE ALIGNMENT

Since the node feature matrices across datasets may differ in dimension, we align them using the following transformation:

$$\hat{x}_i = \mathcal{T}(x_i) \in \mathbb{R}^{d_x}, \quad x_i \in \mathbb{R}^{d_{x_i}} , \tag{2}$$

where $x_i$ denotes the initial feature corresponding to node $v_i$, $d_{x_i}$ and $d_x$ represent the initial attribute dimension and the transformed attribute dimension, respectively. The function $\mathcal{T}(\cdot)$ serves as a transformation operator that aligns feature representations across different datasets. To ensure compatibility with graph data containing numerical attributes, we employ singular value decomposition (SVD) for this transformation.

### 3.2 MULTI-RELATION HYPERNETWORK LEARNING

For each relation token $r$ in the relation vocabulary, we initialize its representation by using a language model (LM). Specifically, we use Sentence-BERT (Reimers, 2019), a widely used model for generating effective sentence representations. The initial relation representation for each token is denoted as:

$$h_r = \text{LM}(r), \quad r \in \mathcal{R}_{\text{source}} \tag{3}$$

#### 3.2.1 AGGREGATOR FOR RELATION MODELING

Inspired by the Relational Graph Convolutional Network (RGCN) (Schlichtkrull et al., 2018), we propose a method that designs tailored aggregators for different relations, accounting for the varying impacts of relationships on nodes. Specifically, REEF adaptively adjusts the message passing mechanism with a hypernetwork that generates relation-specific aggregator parameters from relation representations; that is, the semantic information encoded in the language model is mapped into the aggregator parameters. The mapping can be formalized as follows:

$$\Phi_r^{(l)} = \mathcal{F}_{\text{Agg}}(h_r; \Theta_{r,\text{Agg}}^{(l)}), \quad l = 1, 2 \ldots, L , \tag{4}$$

where $\Phi_r^{(l)}$ denotes the aggregator parameters corresponding to relation token $r$ at the $l$-th layer of the graph neural network. The function $\mathcal{F}_{\text{Agg}}$ is a hypernetwork conditioned on the relation representation $h_r$, and parameterized by $\Theta_{r,\text{Agg}}^{(l)}$. $L$ denotes the total number of layers in the graph neural network. With the parameters of the aggregators obtained, we can perform message passing for different relations within the subgraph. In subgraph $s_i$, the representation $h_{v_i}$ of node $v_i$ at layer $l + 1$ is computed as follows:

$$h_{v_i}^{(l+1)} = h_{v_i}^{(l)} + \sum_{r \in \mathcal{R}_{\text{source}}} \sum_{v_j \in \mathcal{N}_i^r} \frac{1}{|\mathcal{N}_i^r|} \Phi_r^{(l)} h_{v_j}^{(l)} , \tag{5}$$

where $\mathcal{N}_i^r$ represents the set of neighbors of node $v_i$ connected through relation token $r$. To ensure balanced contributions from neighbors, we apply the normalization factor $\frac{1}{|\mathcal{N}_i^r|}$, mitigating the potential over-representation of densely connected nodes. Finally, the representation of node $v_i$ from the last layer, $h_{v_i}^{(L)}$, serves as the overall representation $h_{s_i}$ for the corresponding subgraph $s_i$.

#### 3.2.2 TASK-SPECIFIC CLASSIFIER FOR RELATIONAL INFERENCE

For each training triple $\langle s_i, r_{ij}, s_j \rangle$, we compute the representations of the two subgraphs, $h_{s_i}$ and $h_{s_j}$, respectively. Based on these subgraph representations, REEF utilizes relation token $r_{ij}$ as a supervision signal to predict the type of relationship between the two subgraphs. Since different types of relations correspond to distinct semantics and mapping spaces, we design independent classifiers for each relation to determine whether a specific relation token $r$ exists between the subgraphs. To effectively distinguish various types of relations, we introduce another hypernetwork to adaptively learn the parameters of different classifiers based on the relation representation $h_r$. The relation-specific classifier is defined as follows:

$$\Psi_r = \mathcal{F}_{\text{Cls}}(h_r; \Theta_{r,\text{Cls}}) , \tag{6}$$

where $\mathcal{F}_{\text{Cls}}(h_r; \Theta_{r,\text{Cls}})$ denotes the mapping function that takes the relation representation $h_r$ as input and outputs the classifier parameters $\Psi_r$. Here, $\Theta_{r,\text{Cls}}$ represents the hypernetwork parameters

specific to the classifier for relation token $r$. Subsequently, the representations of subgraphs $h_{s_i}$ and $h_{s_j}$ are combined via the Hadamard product ($\circ$) to form a joint representation $h_{s_{ij}}$. The relation-specific classifier, parameterized by $\Psi_r$, is then applied to predict the probability $P(y \mid \langle s_i, r_{ij}, s_j \rangle)$ of the given relation $r$ between subgraphs $s_i$ and $s_j$, as described by the following equation:

$$h_{s_{ij}} = h_{s_i} \circ h_{s_j}, \ \ P(y \mid \langle s_i, r_{ij}, s_j \rangle) = \text{sigmoid}(\Psi_r h_{s_{ij}}) \in (0,1) \ . \tag{7}$$

Based on the two aforementioned hypernetworks, REEF constructs mapping functions that transform relation tokens into parameters of the aggregators and classifiers. This design allows the graph foundation model to effectively capture a wide range of relations during pretraining, thereby enabling the development of specialized aggregators and classifiers tailored to each specific relation. REEF thus possesses the ability to generalize across diverse relation scenarios.

### 3.3 FEATURE REPRESENTATION ENHANCEMENT

In the process of designing the aggregators and classifiers, we observe that different datasets exhibiting the same relation can share the same aggregator and classifier parameters. However, node attributes across different datasets exhibit significant variations, reflecting the distinct characteristics inherent to each dataset. For example, both Cora and PubMed are citation networks where the edges represent *citation relationships*, and their classification tasks focus on *research papers*. Despite this similarity, the node attributes differ significantly: Cora predominantly contains papers related to *computer science*, while PubMed is centered around *diabetes research*. Such differences make it insufficient to only condition on relations; the distinct characteristics of each dataset must also be captured. To this end, REEF learns a dedicated dataset representation and adapts it through two complementary mechanisms: a dataset-specific projector and a dataset-level feature bias.

Specifically, based on the dataset description $t_\mathcal{G}$ of dataset $\mathcal{G}$, we encode it using Sentence-BERT to obtain its representation $h_{t_\mathcal{G}}$. Here, $h_{t_\mathcal{G}}$ is a learnable embedding that is updated during training, thereby enhancing the flexibility and expressiveness of the representation. Then, we apply the mapping function $\mathcal{F}_{\text{Proj}}$, implemented by the hypernetwork, to generate the projection matrix, which is formulated as follows:

$$h_{t_\mathcal{G}} = \text{LM}(t_\mathcal{G}), \quad \Phi_\mathcal{G}^{(l)} = \mathcal{F}_{\text{Proj}}(h_{t_\mathcal{G}}; \Theta_{\mathcal{G},\text{Proj}}^{(l)}) \ , \tag{8}$$

where $\Phi_\mathcal{G}^{(l)}$ and $\Theta_{\mathcal{G},\text{Proj}}^{(l)}$ represent the parameters of the $l$-th layer, with the former being for the projector and the latter for the hypernetwork. The projection matrix is then applied directly to the representation of the target node $v_i$ at each layer, adapting its hidden state to dataset-specific characteristics. Accordingly, the update of node $v_i$ at layer $l+1$ in the GNN, as given in Eq. 5, is reformulated as:

$$h_{v_i}^{(l+1)} = \Phi_\mathcal{G}^{(l)} h_{v_i}^{(l)} + \sum_{r \in \mathcal{R}_{\text{source}}} \sum_{v_j \in \mathcal{N}_i^r} \frac{1}{|\mathcal{N}_i^r|} \Phi_r^{(l)} h_{v_j}^{(l)} \ , \tag{9}$$

Furthermore, we incorporate a dataset-level feature bias to the initial representation of all nodes in the subgraph. This bias is derived from $h_{t_\mathcal{G}}$ through a transformation function $\mathcal{F}_{\text{Feat}}$ parameterized by $\Theta_{\text{Feat}}$:

$$h_\mathcal{G} = \mathcal{F}_{\text{Feat}}(h_{t_\mathcal{G}}; \Theta_{\text{Feat}}) \in \mathbb{R}^{d_x}, \quad h_v^{(0)} = \hat{x} + h_\mathcal{G}, \ \forall v \in s_i \tag{10}$$

where $\hat{x}$ represents the transformed feature of node $v$ as defined in Eq. 2. This bias term $h_\mathcal{G}$ provides a global dataset-level shift, aligning node features with the broader semantic distribution of the dataset (e.g., shifting toward biomedical semantics in PubMed). Together, these two mechanisms allow REEF to balance local flexibility with global consistency, leading to more robust representation learning across diverse datasets.

In summary, the parameterized hypernetworks in REEF include $\{\Theta_{r,\text{Agg}}, \Theta_{r,\text{Cls}}, \Theta_{\mathcal{G},\text{Proj}}\}$, where $r \in \mathcal{R}_{\text{source}}$. For each relation token $r$, the parameters of its tailored aggregator and classifier, denoted as $\{\Phi_r^{(1)}, \Phi_r^{(2)}, \ldots, \Phi_r^{(L)}\}$ and $\Psi_r$ are computed by two hypernetworks $\mathcal{F}_{\text{Agg}}$ and $\mathcal{F}_{\text{Cls}}$, respectively. The hypernetwork $\mathcal{F}_{\text{Proj}}$ generates the parameters of dataset projector $\{\Phi_\mathcal{G}^{(1)}, \Phi_\mathcal{G}^{(2)}, \ldots, \Phi_\mathcal{G}^{(L)}\}$. Moreover, the feature bias transformation is achieved by learning parameters $\Theta_{\text{Feat}}$ in $\mathcal{F}_{\text{Feat}}$. Building upon this framework, we proceed with the pre-training of the graph foundation model.

### 3.4 PRE-TRAINING STRATEGY

To enhance the robustness and generalization capabilities of REEF, we introduce graph data augmentation and a mixed dataset training strategy during pretraining. These techniques are designed to capture relational diversity more effectively, mitigate overfitting, and enhance adaptability across various graph structures. Specifically, graph data augmentation introduces edge perturbation by randomly removing edges in each training epoch, encouraging the model to accommodate topological variations. In parallel, REEF is trained using a mixed-dataset strategy across multiple graph domains. Each dataset is divided into batches, and all batches are randomly shuffled to construct a unified training queue. This design not only mitigates the risk of overfitting to a single dataset but also balances datasets of varying scales, thereby enhancing generalization.

For downstream tasks, the pretrained foundation model is fine-tuned on the target dataset $\mathcal{G} \in \mathcal{G}_{\text{target}}$. The corresponding aggregators and classifiers are obtained by the learned relation tokens from the relation vocabulary. In addition, the dataset representation $h_{t_{\mathcal{G}}}$ is learned specifically for $\mathcal{G}$.

## 4 EXPERIMENTS

In this section, we present the experimental results of REEF and baseline methods, covering both pre-training and transfer learning. Beyond the main comparisons, we further examine the scaling behavior of REEF under different dataset sizes and evaluate its transferability when initialized with LM-based node features. Additional experimental analyses are provided in the appendix.

### 4.1 EXPERIMENTAL SETTINGS

#### 4.1.1 DATASETS AND BASELINES

**Datasets.** For a fair comparison, we conduct experiments on real-world datasets from four different domains, including (1) two knowledge graphs: FB15K237 and WN18RR (Toutanova & Chen, 2015); (2) three citation networks: Pubmed, Citeseer, and Cora (Sen et al., 2008; Yang et al., 2016); (3) three WebKB subsets: Wisconsin, Texas, and Cornell (Pei et al., 2020); and (4) two Amazon co-purchase networks: Computers and Photo (Shchur et al., 2018). Detailed descriptions of these datasets are provided in Appendix B.

**Baselines.** We compare our proposed framework with three groups of baseline models: (1) Supervised GNN methods including GCN (Kipf & Welling, 2016), GAT (Veličković et al., 2017), and RGCN (Schlichtkrull et al., 2018). (2) Graph contrastive learning methods like GraphCL (You et al., 2020) and SimGRACE (Xia et al., 2022). (3) Graph pre-training methods for transfer: GCOPE (Zhao et al., 2024a) and MDGPT (Yu et al., 2024). For further details, refer to Appendix C.

#### 4.1.2 IMPLEMENTATION DETAILS

REEF includes 7 pre-training datasets: FB15K237, WN18RR, Pubmed, Citeseer, Wisconsin, Texas, and Photo. During pretraining, only the training samples of these datasets are used. For the relation vocabulary, the relation tokens of the aggregators and classifiers for FB15K237 and WN18RR are derived from the edge descriptions in their original datasets. As for Pubmed and Citeseer, they are based on the descriptions of "citation" and "paper classification", respectively. Similarly, for Wisconsin and Texas, the relation token for the aggregator corresponds to the description of "hyperlink", while the relation token for the classifier is based on the description of "web page classification." In the case of Photo, relation tokens are derived from the descriptions of "co-purchase" and "product classification." The transfer learning datasets $\mathcal{G}_{\text{target}}$ include Cora, Cornell, and Computers, whose aggregators and classifiers are consistent with the dataset from the same domain. In total, the size of the relation vocabulary in this pre-training setting is 254.

For evaluation, we assess pre-training performance on the test set of the pre-training datasets $\mathcal{G}_{\text{source}}$, using accuracy (Acc) as the performance metric. In transfer learning, we follow the GCOPE setup, adopting the $C$-way, 1-shot learning, where $C$ represents the number of classes in the target datasets $\mathcal{G}_{\text{target}}$. The remaining data is split into a validation set and a test set in a 1:9 ratio. We run the experiments five times and report the average results. Transfer learning performance is evaluated

Table 1: Comparison of model performance on various pretraining datasets. The table reports accuracy (Acc %) and the performance gap ($\Delta = \text{Top} - \text{Score}$) to the best result on each dataset. The average "Acc" is the mean accuracy calculated across the five datasets presented in the table. The **best** result is bold and underlined, while the runner-up is underlined.

| Domain | Citation | | WebKB | | Amazon | Average | |
|---|---|---|---|---|---|---|---|
| Dataset | **Pubmed** | **Citeseer** | **Wisconsin** | **Texas** | **Photo** | Acc ↑ | Δ ↓ |
| GCN (ind) | **88.42** ($\Delta_{0.00}$) | 76.50 ($\Delta_{0.05}$) | 51.76 ($\Delta_{24.71}$) | 55.14 ($\Delta_{25.81}$) | **93.02** ($\Delta_{0.00}$) | 72.97 | 10.11 |
| GAT (ind) | 86.33 ($\Delta_{2.09}$) | **76.55** ($\Delta_{0.00}$) | 49.41 ($\Delta_{27.06}$) | 52.16 ($\Delta_{28.79}$) | 92.73 ($\Delta_{0.29}$) | 71.44 | 11.65 |
| GCN (joint) | 61.21 ($\Delta_{27.21}$) | 64.86 ($\Delta_{11.69}$) | 61.21 ($\Delta_{15.26}$) | 57.14 ($\Delta_{23.81}$) | 30.45 ($\Delta_{62.57}$) | 54.97 | 28.11 |
| GAT (joint) | 52.93 ($\Delta_{35.49}$) | 57.81 ($\Delta_{18.74}$) | 55.81 ($\Delta_{20.66}$) | 52.38 ($\Delta_{28.57}$) | 28.52 ($\Delta_{64.50}$) | 49.49 | 33.59 |
| RGCN (joint) | 80.30 ($\Delta_{8.12}$) | 71.77 ($\Delta_{4.78}$) | 72.09 ($\Delta_{4.38}$) | **80.95** ($\Delta_{0.00}$) | 79.03 ($\Delta_{13.99}$) | 76.83 | 6.25 |
| REEF | 86.38 ($\Delta_{2.04}$) | 72.37 ($\Delta_{4.18}$) | **76.47** ($\Delta_{0.00}$) | 70.27 ($\Delta_{10.68}$) | 93.01($\Delta_{0.01}$) | **79.70** | **3.38** |

Table 2: Performance comparison of different methods under transfer learning ($C$-way 1-shot). "GCOPE+CL" and "GCOPE+Sim" denote the variants of GCOPE that utilize the contrastive learning frameworks of GraphCL and SimGRACE, respectively. The **best** results are highlighted in bold. "Imporve." represents the improvement of REEF over the best baseline on the corresponding metric.

| Method | Cora (Citation) | | | Cornell (WebKB) | | | Computers (Amazon) | | |
|---|---|---|---|---|---|---|---|---|---|
| | Acc | AUC | F1 | Acc | AUC | F1 | Acc | AUC | F1 |
| GCN | $0.3012_{\pm.06}$ | $0.6444_{\pm.04}$ | $0.2591_{\pm.04}$ | $0.3263_{\pm.04}$ | $0.5666_{\pm.01}$ | $0.3151_{\pm.03}$ | $0.2602_{\pm.07}$ | $0.6773_{\pm.02}$ | $0.2428_{\pm.04}$ |
| GAT | $0.3646_{\pm.04}$ | $0.6769_{\pm.03}$ | $0.3108_{\pm.04}$ | $0.3275_{\pm.14}$ | $0.5306_{\pm.03}$ | $0.1497_{\pm.04}$ | $0.3482_{\pm.07}$ | $0.6878_{\pm.05}$ | $0.2397_{\pm.05}$ |
| GraphCL | $0.2507_{\pm.06}$ | $0.6350_{\pm.03}$ | $0.2240_{\pm.03}$ | $0.4175_{\pm.04}$ | $0.6350_{\pm.02}$ | $0.3500_{\pm.04}$ | $0.2856_{\pm.04}$ | $0.6467_{\pm.03}$ | $0.1653_{\pm.06}$ |
| SimGRACE | $0.2492_{\pm.02}$ | $0.5765_{\pm.03}$ | $0.1567_{\pm.04}$ | $0.3438_{\pm.13}$ | $0.5954_{\pm.09}$ | $0.2168_{\pm.09}$ | $0.2666_{\pm.10}$ | $0.6286_{\pm.01}$ | $0.1603_{\pm.03}$ |
| GCOPE+CL | $0.3368_{\pm.02}$ | $0.6971_{\pm.04}$ | $0.2967_{\pm.03}$ | $0.3975_{\pm.10}$ | $0.6694_{\pm.04}$ | $0.3120_{\pm.04}$ | $0.3439_{\pm.03}$ | $0.7023_{\pm.01}$ | $0.2976_{\pm.03}$ |
| GCOPE+Sim | $0.2525_{\pm.05}$ | $0.5744_{\pm.03}$ | $0.1722_{\pm.06}$ | $0.3675_{\pm.09}$ | $0.6045_{\pm.04}$ | $0.2339_{\pm.04}$ | $0.3230_{\pm.01}$ | $0.6994_{\pm.00}$ | $0.2515_{\pm.00}$ |
| MDGPT | $0.4421_{\pm.08}$ | $0.7912_{\pm.05}$ | $0.4271_{\pm.08}$ | $0.2400_{\pm.08}$ | $0.5776_{\pm.04}$ | $0.1723_{\pm.04}$ | $0.3837_{\pm.09}$ | $0.7950_{\pm.04}$ | $0.4120_{\pm.08}$ |
| **REEF** | **$0.4866_{\pm.04}$** | **$0.8035_{\pm.03}$** | **$0.4800_{\pm.05}$** | **$0.4865_{\pm.08}$** | **$0.7016_{\pm.03}$** | **$0.3649_{\pm.07}$** | **$0.4772_{\pm.06}$** | **$0.8480_{\pm.03}$** | **$0.4403_{\pm.02}$** |
| Improve. | +10.07% | +1.55% | +12.39% | +16.52% | +4.81% | +4.08% | +24.05% | +6.67% | +6.87% |

using classification accuracy (Acc), AUC-ROC (AUC), and F1 score (F1). For further details on the experimental setup and the specific description of relation vocabulary, please refer to Appendix D.

## 4.2 PRE-TRAINING PERFORMANCE

Table 1 presents the pre-training experimental results of REEF compared to other baseline models. For baselines, we consider two training settings. The notation "ind" indicates models trained and tested independently on a single dataset, while "joint" represents models trained simultaneously on five datasets: Pubmed, Citeseer, Wisconsin, Texas, and Photos. In the "joint" setting, all methods utilize the same input features as REEF, and the labels from all datasets are combined into a unified classification space. i.e., the number of output categories is equal to the sum of the categories of all datasets. Furthermore, RGCN employs the same edge type configuration as REEF, where datasets from the same domain share identical edge types.

From the results, we can draw the following key findings: (1) REEF achieves the best overall performance. Among all methods, REEF attains the highest average accuracy of 79.70%, demonstrating superior overall performance and maintaining a leading edge across multiple datasets. (2) REEF demonstrates more stable performance across different datasets, consistently achieving near-optimal or best accuracy. For example, it attains 76.47% on Wisconsin, 70.27% on Texas, and 93.01% on Photo. In contrast, other methods show significant performance fluctuations across different datasets. Notably, REEF has the lowest performance gap ($\Delta$=3.38) among all methods, indicating minimal variance across datasets and the highest overall stability. (3) REEF and RGCN significantly outperform GCN and GAT in the joint training setting, showing that incorporating relation semantics enhances cross-domain generalization. (4) Compared with RGCN, REEF further benefits from its relation vocabulary and dataset-specific hypernetworks, which tailor aggregators and classifiers to relation semantics ($\mathcal{F}_{\text{Agg}}, \mathcal{F}_{\text{Cls}}$) while distinguishing dataset-specific variations via projectors ($\mathcal{F}_{\text{Proj}}$). This design enables both relation-level and dataset-level adaptation, resulting in stronger generalization.

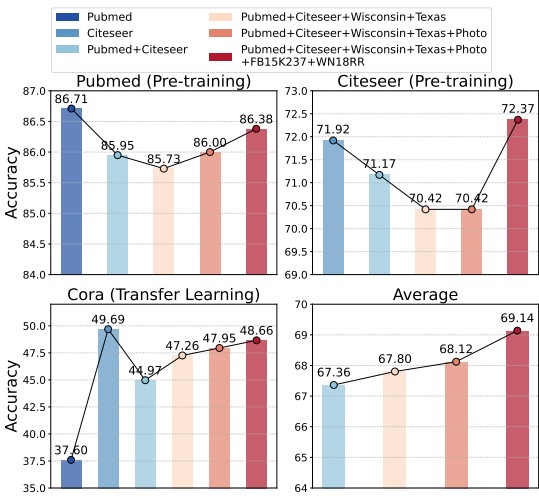

Figure 3: Performance comparison across different dataset scales in the citation domain.

Table 3: 3-shot transfer learning results using LM-based node features. We highlight the best results in **bold**, the second-best in underline, and cross-domain transfer results in *italics*.

| Method | Cora | History | Ratings |
|---|---|---|---|
| GraphMAE | 72.49 | 39.15 | 31.68 |
| LLaGA | 60.70 | 36.45 | 23.45 |
| OFA | 52.49 | 39.36 | 29.08 |
| OFA-FS | 42.10 | 17.50 | 20.50 |
| Prodigy | 40.59 | 19.47 | 20.84 |
| GraphText | 50.33 | 48.00 | 37.67 |
| GraphAdapter | 33.74 | 36.60 | 29.41 |
| GraphLLM | 68.00 | 56.67 | 34.33 |
| GPPT | 44.14 | 27.54 | 14.24 |
| Gprompt | 55.38 | 17.36 | 15.38 |
| SBERT (no pretrain) | 68.42 | 51.25 | 20.95 |
| REEF (Setting 1) | **75.95** | *57.40* | *37.42* |
| REEF (Setting 2) | 74.58 | **62.82** | **38.24** |

In conclusion, REEF demonstrates outstanding generalization, significantly outperforming other methods due to the universal knowledge learned from relation tokens and enhanced dataset-level feature representations.

### 4.3 TRANSFER LEARNING PERFORMANCE

We compare the transfer performance of REEF across datasets from different domains. For graph contrastive learning methods, the backbone is GCN, and the results of the baseline models are directly reported from (Zhao et al., 2024a). The overall results are presented in Table 2. From the table, we observe the following: (1) REEF demonstrates significantly superior transfer performance across all datasets, achieving substantial improvements on all three metrics, with particularly remarkable gains on the Computers dataset, where REEF improves accuracy by 24.05%, AUC by 6.67%, and F1 score by 6.87%. (2) Compared to GCOPE and MDGPT that treat the dataset level as the fundamental unit, REEF places greater emphasis on predicting relationships between nodes, thereby enabling more coherent and effective knowledge transfer across domains. (3) These results underscore the critical role of relation tokens in improving both the generalizability and transferability of graph foundation models.

### 4.4 SCALING LAWS IN PERFORMANCE ACROSS DIFFERENT DATASET SCALES

We examine REEF under six pre-training settings covering different dataset scales (see Figure 3), and evaluate pre-training on Pubmed and Citeseer, transfer learning on Cora, and average performance across datasets. The key observations are summarized as follows:

(1) **Pre-training performace:** Pubmed and Citeseer, when used individually as pre-training datasets, both exhibit strong downstream performance. As more datasets are added, performance may fluctuate due to distributional shifts in multi-dataset training, but it consistently improves as the dataset scale grows. This rebound stems from an enriched vocabulary and exposure to diverse relations, which help the model learn more generalizable representations. This further emphasizes the importance of the relation vocabulary, as different relations play a crucial role in the graph foundation model's ability to learn universal knowledge.

(2) **Transfer learning performance:** Pre-training on Citeseer alone achieves the best transfer accuracy on Cora, likely due to topological and semantic similarities. In contrast, although Pubmed is also a citation network, it shows the worst transfer performance. The reason is that Citeseer and Cora both belong to the computer science field, making their aggregators and classifiers more similar, whereas Pubmed, from the medical field, differs more from Cora. Consequently, the combination of Pubmed and Citeseer leads to inferior transfer performance compared to using Citeseer

alone. Nevertheless, as the training dataset scale increases, the performance of Cora keeps improving. This phenomenon shows that incorporating training data with diverse relations can enhance the model's transferability, leading to better cross-dataset generalization.

(3) **Overall average performance:** As the dataset scale increases and relations become more enriched, the average accuracy gradually improves. Although certain tasks may experience temporary performance drops, expanding the dataset scale generally provides richer representation learning signals, ultimately benefiting the model. The observed results further validate the effectiveness of REEF as a pre-training framework for graph foundation models, while also demonstrating a certain degree of scaling law, indicating that increasing the training dataset scale enhances the model's generalizability and transferability.

### 4.5 Transfer Learning with LM-based Node Features

In the main experiments, we focus on node classification where node features are expressed as numerical attributes. This setting is broadly applicable, as numerical representations provide a unified format for heterogeneous node properties. Nevertheless, it does not adequately reflect scenarios in text-attributed graphs (TAGs), where node semantics are inherently encoded in textual descriptions. To examine the cross-feature generalization capacity of REEF, we conduct a case study in which node features are derived from language model (LM). Following the baseline and evaluation protocol of TSGFM (Chen et al., 2024c), a benchmark framework for Text-space Graph Foundation Models, we evaluate REEF in a 3-shot transfer learning setup with only three labeled samples per class. The transfer evaluation covers three datasets: Cora from the citation network domain, and History and Ratings from the E-commerce domain. REEF is pretrained under two configurations: (i) *Setting 1*, which utilizes FB15K237, WN18RR, Pubmed, and Citeseer; and (ii) *Setting 2*, which extends Setting 1 with the additional inclusion of the Photo dataset. Further details on datasets and baselines are provided in Appendix E.

As shown in Table 3, REEF demonstrates consistently strong transferability across text-attributed graphs, highlighting its generalization capabilities and robustness. Beyond this, REEF outperforms other baselines specifically designed for TAGs, such as OFA, GraphText, and GraphLLM. For example, REEF attains 75.95% on Cora and 62.82% on History, representing substantial gains over the strongest baselines (72.49% and 56.67%, respectively). Notably, in Setting 1, the transfer datasets History and Ratings belong to the E-commerce domain, which is entirely absent from the pretraining corpora. Although REEF under Setting 1 performs slightly lower than Setting 2 on these datasets, it still surpasses almost all competing baselines, further underscoring REEF's strong capability in cross-domain transfer.

Due to space limitations, additional results and analyses are deferred to the Appendix, we defer additional results and analyses to the Appendix, including: visualization of learned representations (Appendix F), computational complexity analysis (Appendix G), performance on link prediction tasks (Appendix H.1), ablation studies (Appendix H.2), out-of-domain relation evaluation transfer performance on out-of-domain relations (Appendix H.3), the impact of varying hidden dimensions (Appendix H.4), and pre-training performance trends across different datasets (Appendix H.5).

## 5 Conclusion

In this paper, we proposed REEF, a novel framework that introduces relation tokens as fundamental units for graph foundation models. We constructed a relation vocabulary to store relational information and design two hypernetworks to adaptively generate the parameters of aggregators and classifiers based on relation tokens. Additionally, another hypernetwork generates dataset-specific feature projectors and incorporate a dataset-level feature bias based on dataset description, improving flexibility across datasets with the same relations. By leveraging graph data augmentation and a mixed-dataset pretraining strategy, REEF effectively captures relational diversity and improves generalization. Comprehensive experiments and analyses demonstrate that REEF performs exceptionally well in both pretraining and transfer learning tasks, exhibiting strong generalizability and transferability while also revealing scaling laws to some extent. These findings highlight the pivotal role of relations as fundamental units, suggesting that relation-centric design is essential for advancing the next generation of graph foundation models.

ETHICS STATEMENT

Our proposed graph foundation model is developed and evaluated solely on publicly available benchmark datasets that contain no personally identifiable information, and all usage follows the original licenses. We are aware that graph foundation models may inherit and amplify biases present in the data; thus, we report dataset statistics, publish variance across runs, and encourage further audits before deployment in sensitive applications. While the model has the potential to support a wide range of downstream tasks, it is released strictly for research purposes and not intended for high-stakes decision-making without expert oversight and compliance checks. We also report compute usage and adopt efficiency practices to mitigate environmental impact. No human-subjects research was conducted, and our code and scripts are released to ensure transparency and reproducibility.

REPRODUCIBILITY

To support reproducibility, we provide detailed implementation instructions for REEF in Appendix D, and release the complete source code at `https://anonymous.4open.science/r/REEF-16CF/`. These resources enable other researchers to readily verify and replicate our results.

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

## A RELATED WORK

With the remarkable success of LLMs, GFMs have emerged as a prominent research direction in the graph domain (Mao et al.; Dong et al., 2024; Liu et al., 2023b), aiming to improve generalization across tasks and domains through training on large-scale and diverse graph datasets (Mao et al.; Dong et al., 2024; Liu et al., 2023b). To explore this emerging paradigm, existing studies can be broadly categorized into two main lines of research: those that leverage LLMs as the backbone for GFMs, and those that rely on GNNs to construct GFMs.

**LLM-based GFMs.** Some methods investigate their feasibility by leveraging LLMs (Zhu et al., 2024b; Tan et al., 2024; Chen et al., 2024a; Wang et al., 2024a). For example, MuseGraph unifies graph descriptions, chain-of-thought generation, and graph-aware instruction tuning to enable LLMs on diverse graph mining tasks. LLaGA introduces lightweight projectors to align graph embeddings with LLM representations. GOFA (Kong et al., 2024) and ZeroG (Li et al., 2024) adapt language models through fine-tuning to enhance their transferability in zero-shot settings. These approaches benefit from the powerful language understanding of LLMs, but they often struggle with structural complexity in graphs—particularly as graph size grows or sequence length increases.

**GNN-based GFMs.** Another line of work focuses on utilizing GNNs to build GFMs (Zhao et al., 2024b). Some methods generalize pre-trained models across tasks via graph prompts (Zhu et al., 2024a; Gong et al., 2024; Sun et al., 2023; Fang et al., 2024; Liu et al., 2023c;a). Others empha-size the generalization capability across different datasets, where various approaches utilize distinct graph elements as base units to learn how GFMs generalize to different datasets or domains. For node-level generalization, Prodigy (Huang et al., 2024a) constructs task graphs that propagate label information from the support set to queries, thereby enabling in-context learning. OFA (Liu et al., 2023a) introduces the nodes-of-interest (NOI) subgraph and the NOI prompt node, enabling joint training across multiple datasets and unifying different types of graph tasks. OpenGraph (Xia et al., 2024) develop a graph tokenizer that transforms input graphs into unified token sequences, with each token representing a node and a semantic vector, thereby standardizing node distributions across different graphs. For dataset-level generalization, OMOG (Liu et al., 2024) and AnyGraph (Xia & Huang, 2024) both adopt Mixture-of-experts (MoE) (Cai et al., 2024), training a separate model for each pretraining graph to capture its unique structural and feature characteristics. GCOPE (Zhao et al., 2024a) introduces coordinators to merge multiple graph datasets into a single large graph

during pretraining, facilitating knowledge transfer across datasets. Moreover, MDGPT (Yu et al., 2024) employs domain tokens to align graph features across different domains and incorporates a dual prompt mechanism to further enhance adaptation to target tasks.

Despite these advances, most existing approaches rely on node-level or dataset-level pretraining, which restricts flexibility and scalability. In contrast, REEF introduces relation tokens as the fundamental unit for pre-training. This design allows REEF to unify diverse relations across datasets, while dataset-specific projectors and biases adapt to distributional differences. As a result, REEF not only supports comprehensive pre-training but also enables more effective fine-tuning and transfer, thereby enhancing generalization to new domains.

## B DATASETS

Table 4 summarizes the details. Here is a detailed description of ten datasets in four domains:

- **Knowledge graphs:**
    - FB15K237 is a knowledge graph containing relation triples and textual mentions of Freebase entity pairs. The raw text data for the nodes was collected from a GitHub repository[1]. The text feature of a node includes the entity name and description, while the text feature of an edge describes the relation type between entities.
    - WN18RR is a subset of WordNet, consisting of 11 relations and 40,943 entities. The raw text data for WN18RR nodes is also collected from the same GitHub repository[2]. Node and edge text features are processed in the same way as FB15K237.
- **Citation networks:** Cora, Citeseer, and Pubmed are widely used citation network benchmarks. In these networks, nodes represent papers, and edges denote citations of one paper by another. Node features are represented as bag-of-words vectors of the paper content, and node labels correspond to the paper's research domain.
    - Cora is a citation network in the computer science domain, where nodes represent papers and edges denote citation relationships. The dataset is categorized into seven research areas: case-based, genetic algorithms, neural networks, probabilistic methods, reinforcement learning, rule learning, and theory.
    - PubMed is a citation network of biomedical papers, categorized into three themes: diabetes, experimental diabetes, and type 1/type 2 diabetes. Edges indicate citation relationships between papers.
    - CiteSeer is a citation network consisting of academic papers and their citation relationships in the computer science domain. The dataset is categorized into six research fields: agents, artificial intelligence, database, information retrieval, machine learning, and human–computer interaction.
- **WebKB**[3]**:** The WebKB dataset was collected by Carnegie Mellon University from computer science departments of various universities, consisting of three sub-datasets: Cornell, Texas, and Wisconsin. In these datasets, nodes represent web pages, and edges correspond to hyperlinks between them. The feature vectors of the nodes are derived from the bag-of-words representation of the web pages. These web pages are manually classified into five categories: student, project, course, staff, and faculty.
- **Amazon:** Amazon Computers and Amazon Photo are segments of the Amazon co-purchase graph, where nodes represent goods, edges indicate that two goods are frequently bought together, node features are bag-of-words encoded product reviews, and class labels are given by the product category.

## C BASELINES

We compare our proposed framework with three groups of baseline models: (1) General supervised GNN methods including GCN (Kipf & Welling, 2016), GAT (Veličković et al., 2017), and

---

[1]`https://github.com/villmow/datasets_knowledge_embedding/tree/master`
[2]`https://github.com/villmow/datasets_knowledge_embedding/tree/master`
[3]`http://www.cs.cmu.edu/afs/cs.cmu.edu/project/theo-11/www/wwkb`

Table 4: Statistics of datasets.

| Dataset | Domain | #Nodes | #Edges | #Classes | Task | Type |
|---|---|---|---|---|---|---|
| FB15K237 | Knowledge | 14,541 | 310,116 | 237 | Link | $\mathcal{G}_{source}$ |
| WN18RR | Knowledge | 40,943 | 93,003 | 11 | Link | $\mathcal{G}_{source}$ |
| PubMed | Citation | 19,717 | 44,338 | 3 | Node | $\mathcal{G}_{source}$ |
| Citeseer | Citation | 3,237 | 9,104 | 6 | Node | $\mathcal{G}_{source}$ |
| Cora | Citation | 2,708 | 10,556 | 7 | Node | $\mathcal{G}_{target}$ |
| Wisconsin | WebKB | 251 | 515 | 5 | Node | $\mathcal{G}_{source}$ |
| Texas | WebKB | 183 | 325 | 5 | Node | $\mathcal{G}_{source}$ |
| Cornell | WebKB | 183 | 298 | 5 | Node | $\mathcal{G}_{target}$ |
| Photo | Amazon | 7,650 | 238,162 | 8 | Node | $\mathcal{G}_{source}$ |
| Computers | Amazon | 13,752 | 491,722 | 10 | Node | $\mathcal{G}_{target}$ |

RGCN (Schlichtkrull et al., 2018). These models rely on task-specific supervision and serve as fundamental GNNs for graph representation learning. (2) Graph contrastive learning methods like GraphCL (You et al., 2020) and SimGRACE (Xia et al., 2022), which leverage contrastive objectives to learn graph representations in a self-supervised manner. The learned representations are then fine-tuned on downstream tasks to improve task-oriented performance. (3) Graph pre-training methods for transfer: GCOPE (Zhao et al., 2024a) employs a pretraining framework for contrastive learning that integrates information across multiple datasets through coordinators, enabling knowledge transfer to downstream tasks. MDGPT (Yu et al., 2024) leverages domain tokens to align graph representations from different domains and adopts a dual-prompt strategy to further improve adaptation to the target task.

## D EXPERIMENTAL SETTINGS

In our proposed framework REEF, we implement it with Pytorch and adopt the Adam optimizer for training. For data splitting during the pretraining stage, FB15K237 and WN18RR are split according to 142, 47, 48, Wisconsin and Texas follow their original splits, while Pubmed, Citeseer, and Photo are divided using a 6:2:2 ratio. To unify feature representations across different datasets, we apply Singular Value Decomposition (SVD) to reduce the initial feature dimensions to 128. We fix the number of layers of the graph neural network $L$ at 2 for REEF and other baselines. For REEF, we set the learning rate to 0.0002, the dropout rate to 0.5, the hidden dimension to 64, batch size to 128, and train for 100 epochs. Each of the three hypernetworks $\mathcal{F}_{Agg}, \mathcal{F}_{Cls}, \mathcal{F}_{Proj}$ and the feature bias transformation function $\mathcal{F}_{Feat}$ adopts a two-layer MLP structure. The edge mask rate in graph data augmentation is set to 0.2. For fairness, we run all the experiments on a server with 80G memory and a single Tesla A800 GPU. For a detailed description of the relations, see Table 8.

For the baselines in transfer learning, we follow the original experimental settings of GCOPE (Zhao et al., 2024a) and MDGPT (Yu et al., 2024), respectively. Specifically, GCOPE involves ten datasets; following its protocol, we pretrain the model on nine datasets and conduct transfer evaluation on the remaining one. Similarly, MDGPT involves seven datasets, where the model is pretrained on six datasets and evaluated on the remaining one. Since the Cornell dataset is not included among the seven datasets used in MDGPT, we pretrain on these seven datasets and evaluate the transfer performance on Cornell.

## E EXPERIMENTAL DETAILS FOR LM-BASED NODE FEATURES

While the main experiments of this paper focus on graphs with numerical node attributes, such a setting does not fully cover scenarios where node semantics are expressed in natural language. To complement the main results, we further evaluate REEF under a text-attributed graph setting, where node features are derived from language-model (LM) embeddings of textual attributes. This case study is intended to examine whether REEF's relation-centric pretraining generalizes across feature modalities, following the evaluation design of TSGFM (Chen et al., 2024c).

**Datasets.** To complement the numerical-feature setting discussed in the main experiments, here we provide details of the text-attributed graphs (TAGs) used in the LM-based case study. In this

setting, node features are derived from textual descriptions, which are encoded into embeddings by Sentence-BERT (Reimers, 2019). Comprehensive dataset statistics are available in the official TSGFM repository[4].

- **Citation Networks:** In these graphs, nodes represent papers and edges denote citation links. Unlike the numerical-feature setting, where bag-of-words vectors are commonly used, here each node is associated with textual descriptions of the publication (e.g., title, abstract, and keywords), while edges may also carry textual annotations describing the citation context.
  - **Cora:** Contains 2,708 scientific publications from the computer science domain, categorized into 7 classes. Node text features are derived from the paper titles and abstracts, and edges represent citation relationships.
  - **CiteSeer:** Consists of 3,186 publications across 6 categories. Each node is enriched with textual metadata (e.g., title and subject area), while edges denote citation links among papers.
  - **Pubmed:** A biomedical citation network with 19,717 scientific papers in 3 categories. Node text features correspond to paper abstracts focused on diabetes-related research, and edges capture citation connections.
- **E-commerce Graphs:** Nodes in these graphs correspond to products, and their textual attributes include item names, categories, and user-provided descriptions. Edges reflect relations such as co-purchase links or user ratings, and thus the graph structure is tightly coupled with semantic product information.
  - **History:** Contains 41,551 products and 503,180 co-purchase edges. Each node has textual attributes describing product titles and categories, covering 12 product classes.
  - **Photo:** Comprises 48,362 electronic products with 873,782 co-purchase edges. Node-level text features consist of product descriptions, with items grouped into 12 categories. Note that this TAG version differs in scale from the numerical-feature Photo dataset used in the main experiments.
  - **Amazon Ratings:** A user–item bipartite graph containing 24,492 nodes and 186,100 rating edges. Product nodes are annotated with textual descriptions, and the dataset spans 5 product categories.

**Baselines.** We compare REEF against a comprehensive set of representative models from TS-GFM (Chen et al., 2024c), covering different paradigms of text-space graph foundation models: (1) *Graph self-supervised learning (SSL):* GraphMAE (Hou et al., 2022) reconstructs masked node features to learn transferable graph representations. (2) *Foundational graph prompt models:* GPPT (Sun et al., 2022) and Gprompt (Liu et al., 2023c) unify pretext and downstream tasks as edge prediction. Prodigy (Huang et al., 2024a) reformulates classification tasks as link prediction over prompt graphs, thereby enabling in-context learning on graph data. OFA (Liu et al., 2023a) introduces the nodes-of-interest (NOI) subgraph and the NOI prompt node, enabling joint training across multiple datasets and unifying different types of graph tasks. (3) *LLM-based methods:* These approaches project graph data into the text space to align with pretrained language models. GraphLLM (Chen et al., 2024b) and GraphText (Zhao et al., 2023) directly leverage LLMs to perform graph classification and reasoning tasks. GraphAdapter (Huang et al., 2024b) adopt GPT2 [88] as the backbone LLMs considering the computation resource restriction. LLaGA (Chen et al., 2024a) employs lightweight projectors to align graph embeddings with the LLM space. (4) *Lightweight baseline:* SBERT (no pretrain) directly uses sentence embeddings from Sentence-BERT (Reimers, 2019) as node features without graph pretraining.

## F  VISUALIZATION OF REPRESENTATIONS

The t-SNE visualizations in Figure 4 illustrate the distribution of representations across different datasets after pre-training. We can see that: (1) The pretraining datasets ($\mathcal{G}_{\text{source}}$) form well-structured and distinct clusters. Datasets within the same domain are positioned more closely, while datasets from different domains are farther apart, demonstrating the model's ability to differentiate between different types of data, which is crucial for generalization. Moreover, different relations exhibit a certain degree of separation, with knowledge graph datasets clustering closely together and

---

[4] https://github.com/CurryTang/TSGFM

citation datasets forming distinct groups. (2) Before fine-tuning, the target datasets ($\mathcal{G}_{\text{target}}$) are projected into the pretraining space. Notably, the representations of the Cora and Computers datasets exhibit strong internal clustering and align closely with their corresponding source-domain clusters, reflecting consistency in the learned relational structures. These findings suggest that the pre-trained model effectively captures meaningful relational patterns and exhibits strong feature transferability, providing a solid foundation for downstream tasks.

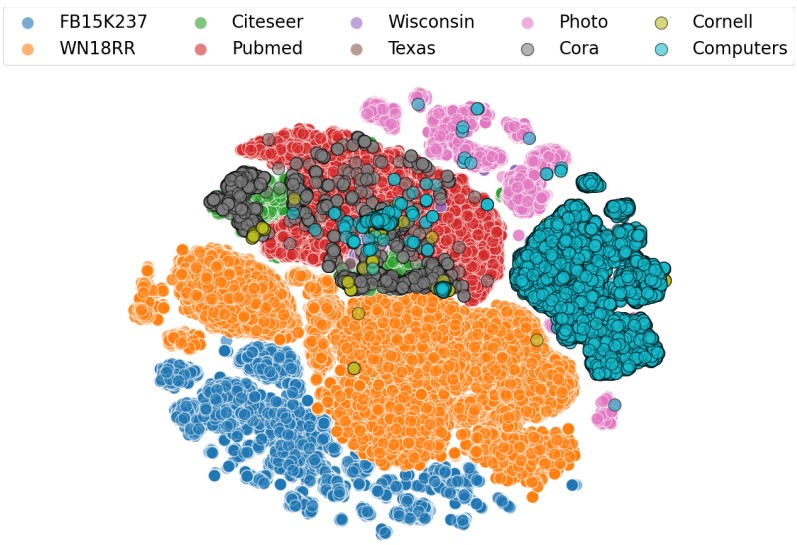

Figure 4: T-SNE visualization of representations after pretraining.

## G  COMPLEXITY ANALYSIS

For our proposed framework REEF, the parameters are $\{\Theta_{r,\text{Agg}}, \Theta_{r,\text{Cls}}, \Theta_{\mathcal{G},\text{Proj}}, \Theta_{\text{Feat}}\}$, corresponding to three hypernetworks $\{\mathcal{F}_{\text{Agg}}, \mathcal{F}_{\text{Cls}}, \mathcal{F}_{\text{Proj}}\}$ and the feature bias transformation function $\mathcal{F}_{\text{Feat}}$. Each of these four functions is implemented as a two-layer MLP, The computational complexity of one such function is $O(d^3)$, where $d$ represents the hidden dimension. Thus, the total complexity contributed by these functions is: $O(L \cdot |\mathcal{R}_{\text{source}}| \cdot d^3 + |\mathcal{R}_{\text{source}}| \cdot d^3 + L \cdot |\mathcal{G}_{\text{source}}| \cdot d^3 + |\mathcal{G}_{\text{source}}| \cdot d^2)$, which simplifies to $O(L \cdot (|\mathcal{R}_{\text{source}}| + |\mathcal{G}_{\text{source}}|) \cdot d^3)$. Here, $L$ denotes the number of layers in the GNN. $|\mathcal{R}_{\text{source}}|$ and $|\mathcal{G}_{\text{source}}|$ represent the size of relation vocabulary and the number of pre-training datasets, respectively. Then, for GNN components, the complexity is $O(L \cdot (|\mathcal{E}_{\mathcal{G}_{\text{source}}}| \cdot d + |\mathcal{R}_{\text{source}}| \cdot |\mathcal{V}_{\mathcal{G}_{\text{source}}}| \cdot d^2))$, where $|\mathcal{E}_{\mathcal{G}_{\text{source}}}|$ denote the total number of edges, and $|\mathcal{V}_{\mathcal{G}_{\text{source}}}|$ represents the total number of nodes in the pre-training datasets. Combining these, the overall complexity of the framework is $O\left(L \cdot \left((|\mathcal{R}_{\text{source}}| + |\mathcal{G}_{\text{source}}|) \cdot d^3 + |\mathcal{E}_{\mathcal{G}_{\text{source}}}| \cdot d + |\mathcal{R}_{\text{source}}| \cdot |\mathcal{V}_{\mathcal{G}_{\text{source}}}| \cdot d^2\right)\right)$.

In summary, compared to RGCN, the time complexity of our proposed framework includes two additional components: the relation vocabulary and the feature enhancement modules, which are related to the size of the relation vocabulary and the number of datasets. The main factor influencing complexity remains the data scale, including the number of nodes and edges in the pre-training datasets. Overall, our framework achieves a favorable complexity, making it well-suited for graph foundation models and scalable for large-scale graph data training.

## H  SUPPLEMENTARY EXPERIMENTS

As a complement to Section 4, we provide additional experimental results and analyses in this section.

- RQ1: How does REEF perform on downstream link prediction tasks?
- RQ2: How do the main components of REEF contribute to its overall performance?

- RQ3: How does the REEF framework perform when encountering an out-of-domain relation?

- RQ4: What is the impact of the hidden dimensions affect the performance of the framework?

- RQ5: How does the performance vary across differenet datasets perform during pre-training?

## H.1 LINK PREDICTION (RQ1)

We evaluate the performance of REEF on the downstream task of link prediction. Relevant baseline models are categorized into three groups: (1) Supervised GNNs and MLP: This group includes a basic MLP (Linear), GCN (Kipf & Welling, 2016), GAT (Veličković et al., 2017), and GIN (Xu et al., 2018). (2) Self-supervised GNNs: DGI (Veličković et al., 2018) leverages contrastive learning between global graph summaries and local node patches. BGRL (Thakoor et al., 2021) adopts bootstrapping to predict representations of the same node across different augmented views. Graph-MAE (Hou et al., 2022) reconstructs node features based on structural information. GIANT (Chien et al., 2021) integrates pretrained language models with GNNs under a self-supervised learning paradigm. (3) Graph Foundation Model: GFT (Wang et al., 2024b) builds a GFM by treating computation trees as transferable vocabulary units to promote generalization and mitigate negative transfer. Here, GCOPE (Zhao et al., 2024a) and MDGPT (Yu et al., 2024) are not used because their original papers do not include link prediction experiments. For fairness, the baseline results are reported from (Wang et al., 2024b), and REEF is evaluated under the same data split setup.

Table 5 presents the accuracy results of various models evaluated on the link prediction task across two benchmark datasets, FB15K237 and WN18RR. Among supervised and self-supervised baselines, performance varies moderately; for instance, GFT achieves strong results with 89.72% accuracy on FB15K237 and 91.91% on WN18RR. Building upon this foundation, REEF achieves the best performance across both datasets, with 91.04% on FB15K237 and 94.70% on WN18RR, significantly outperforming all baselines. These results highlight REEF's superior generalization ability and its effectiveness in link prediction tasks across diverse relational graphs.

Table 5: Accuracy results on the link prediction task.

| Dataset | Linear | GCN | GAT | GIN | DGI | BGRL | GraphMAE | GIANT | GFT | **REEF** |
|---------|--------|-----|-----|-----|-----|------|----------|-------|-----|----------|
| FB15K237 | 87.39 | 82.22 | 88.93 | 83.21 | 81.34 | 80.66 | 85.30 | 87.45 | 89.72 | **91.04** |
| WN18RR | 78.50 | 73.79 | 80.16 | 74.02 | 75.75 | 75.44 | 78.99 | 84.36 | 91.91 | **94.70** |

Table 6: Ablation Study on REEF. "Pre." refers to pretraining, "Trans." refers to transfer learning, and "All" denotes the overall average performance. We highlight the best score in bold.

| Method | Pre-training datasets | | | | | | | Transfer datasets | | | Avg. | | |
|--------|----------|---------|--------|----------|-----------|-------|-------|-------|---------|-----------|--------|--------|--------|
| | FB15K237 | WN18RR | Pubmed | Citeseer | Wisconsin | Texas | Photo | Cora | Cornell | Computers | Pre. | Trans. | All |
| REEF-LM | **91.33** | 94.06 | 86.05 | 70.42 | 64.71 | 59.46 | **93.53** | 46.58 | 44.72 | 43.46 | 79.94 | 44.92 | 69.43 |
| REEF-FB | 89.90 | 92.18 | 86.23 | 70.27 | 70.59 | 54.05 | 92.16 | 40.07 | 44.51 | 42.09 | 79.34 | 42.22 | 68.21 |
| REEF-FP | 90.72 | 94.29 | 85.04 | 73.12 | 56.86 | 59.46 | 93.33 | 40.22 | 43.69 | 46.50 | 78.97 | 45.10 | 68.32 |
| REEF-AGU | 91.14 | 93.78 | **86.41** | 71.62 | 68.63 | 64.86 | 92.22 | 40.81 | 43.69 | 46.68 | 81.24 | 43.73 | 69.98 |
| REEF | 91.04 | **94.70** | 86.38 | **72.37** | **76.47** | **70.27** | 93.01 | **48.66** | **48.65** | **47.72** | **83.46** | **48.34** | **72.93** |

## H.2 ABLATION STUDY (RQ2)

We conduct an ablation study on REEF to better understand the contribution of its main components. To assess the effectiveness of the relational representations in the relation vocabulary, we initialize the relational representations randomly, rather than obtaining them from LM, referring to this variant as REEF-LM. To investigate whether feature augmentation across different datasets under the same relations can further enhance the model's capabilities, we train two variants without the feature bias and feature projector, named REEF-FB and REEF-FP, respectively. Moreover, to evaluate the impact of graph data augmentation on the model's generalization, we use a variant without graph data augmentation, called REEF-AGU. By the way, we have validated the combined effect of the tailored aggregators, classifiers, and feature representation enhancement in our experiments. By comparing REEF with RGCN, the results demonstrate that the hypernetwork design enables the model to learn both the specific relational characteristics of each relation token and the dataset-specific distribution.

Table 6 presents the results. We can observe that: (1) REEF achieves the highest average score on pre-training datasets. Among the variants, REEF-LM performs the worst, indicating the crucial role of semantic information in relational representation initialization. REEF-FB and REEF-FP perform worse overall, indicating that the model lacks the ability to distinguish finer-grained variations across datasets. REEF-AGU also shows a moderate decline, indicating that graph augmentation enhances the model's generalizability. (2) On transfer datasets, REEF again achieves the best performance, demonstrating its superior transferability. REEF-LM again performs relatively poorly, further confirming that the lack of LM-based relational representations, which provide crucial semantic information, leads to weaker knowledge transfer. Removing feature prompts leads to a slight drop, reinforcing their role in improving transfer learning. The absence of graph augmentation results in a decline, suggesting that while augmentation aids generalization. (3) Overall, REEF consistently surpasses its ablated counterparts, demonstrating that LM-based relation embeddings, feature prompts, and graph augmentation are all indispensable for achieving strong generalization and transferability.

Table 7: Transfer Learning Performance on Out-of-Domain Relations for Cornell and Computers Datasets.

| Pre-training datasets | Cornell (#Labels = 5) | | | Computers (#Labels = 10) | | |
|---|---|---|---|---|---|---|
| | Acc | AUC | F1 | Acc | AUC | F1 |
| Pubmed+Citeseer+Wisconsin+Texas+ Photo+FB15K237+WN18RR (REEF) | 48.65±7.76 | 70.16±2.82 | 36.49±6.77 | 47.72±5.61 | 84.80±3.22 | 44.03±2.49 |
| Pubmed+Citeseer+FB15K237+WN18RR | 34.58±12.87 | 57.45±4.57 | 20.29±6.34 | 26.40±11.19 | 47.65±6.20 | 6.10±1.35 |

### H.3 Transfer Performance on Out-of-Domain Relations (RQ3)

In this experiment, we perform transfer learning on out-of-domain relationships on Cornell and Computers. During fine-tuning, we separately introduce the relations corresponding to the Cornell dataset, including "hyperlinks" and "webpage classification", and those from the Computers dataset, including "co-purchase" and "product classification." The pre-training datasets include multiple datasets, including Pubmed, Citeseer, FB15K237, and WN18RR, which include both citation and knowledge graph domains. As shown in Table 7, we observe the following:

(1) Effectiveness of Fine-tuning and Generalization: Since Cornell has 5 labels and Computers has 10, the results show that fine-tuning is effective across these datasets. This demonstrates that even when the pre-training data does not contain relationships specifically corresponding to the out-of-domain relations in the target dataset, our proposed framework REEF can still effectively transfer knowledge and adapt to new, unseen relations, exhibiting strong generalization ability.

(2) Importance of Relations in the Transfer Dataset: The effectiveness of transfer learning is closely tied to the quality and diversity of the relationship data in the pre-training dataset. The more diverse the relations in the pre-training dataset, the better the model performs in transfer learning tasks. By comparing different pre-training configurations, we find that pre-training with a dataset containing a wider variety of relations leads to significantly better performance on the target tasks. For instance, the pre-training configuration involving Pubmed, Citeseer, Wisconsin, Texas, FB15K237, and WN18RR outperforms another configuration with less pre-training datasets in three metrics, demonstrating the importance of rich relational data in pre-training.

These experimental results indicate that the proposed framework REEF has strong adaptability when handling transfer learning tasks, particularly when faced with new or unseen relationships, allowing it to achieve good performance transfer. Additionally, the diversity of relationships in the pre-training dataset plays a crucial role in enhancing the model's performance.

### H.4 Impact of Hidden Dimensions on Model's performance (RQ4)

We set the hidden dimension in our proposed framework REEF to 8, 16, 32, and 64 to compare the model's performance. In Figure 5, we can see that as the hidden dimension increases, most datasets exhibit an upward trend in accuracy, and the overall average performance (Avg) also improves. This indicates a scaling law in REEF, where larger model capacity and increased parameter size generally lead to better overall performance. For smaller datasets such as Wisconsin and Texas, model performance is more sensitive to the hidden dimension, resulting in greater fluctuations. This

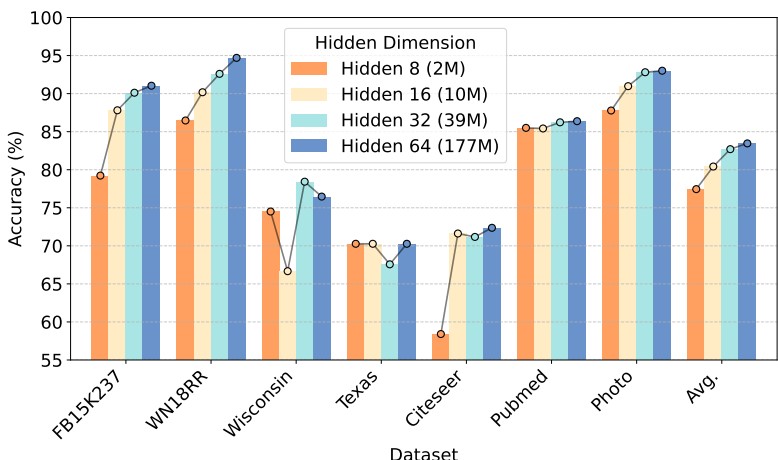

Figure 5: Performance of different datasets with different hidden dimensions. Model size is indicated in parentheses.

instability may stem from the high variance caused by the small dataset size, leading to differences in generalization ability across different hidden dimensions.

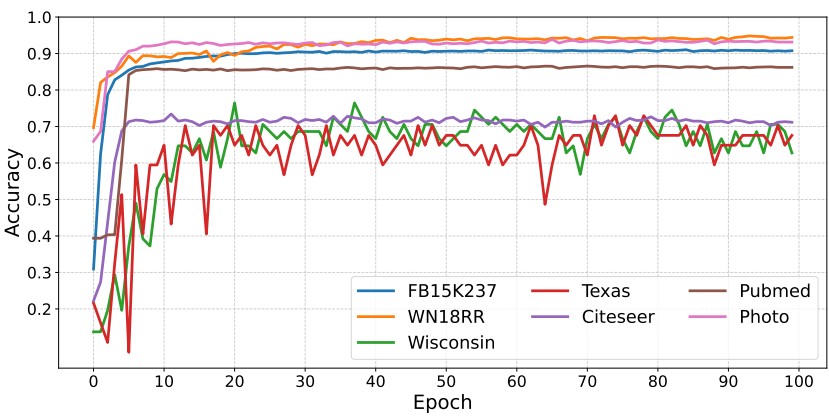

Figure 6: Test accuracy of different datasets during pre-training.

## H.5 PERFORMANCE TRENDS ACROSS DIFFERENT DATASETS DURING PRE-TRAINING (RQ5)

Figure 6 presents the test accuracy trends of the framework REEF across multiple datasets during the pre-training process. Overall, the test accuracy of all datasets increases rapidly within the first 10 epochs and then stabilizes, indicating that the model can quickly learn effective features in the early training stages. Among them, FB15K237, WN18RR, Photo, and Pubmed achieve high accuracy early and maintain stability throughout training, demonstrating good convergence properties. Notably, the Photo dataset stabilizes around 0.9, likely due to the relatively simple nature of its task and the large size of its training data. In contrast, the test accuracy of the Wisconsin and Texas datasets exhibits significant fluctuations, with final accuracy around 0.6-0.7, indicating weaker generalization ability on these datasets, potentially due to their small scale. Additionally, the Citeseer dataset stabilizes around 0.7, showing more stability than Wisconsin and Texas but still underperforming compared to FB15K237 and WN18RR. Regarding convergence stability, the error curves of Photo, Pubmed, and WN18RR are relatively smooth during training, suggesting that the model is more stable on these datasets, likely due to their structured data distributions or the model's strong learning capabilities for these types of data. In contrast, Wisconsin and Texas show greater fluctuations,

especially Texas, indicating less stable training on these datasets, possibly due to their small size or complex distributions, making the model prone to overfitting or sensitive to random variations.

Overall, the pre-trained model exhibits significant performance differences across datasets, excelling on structured knowledge graphs (FB15K237, WN18RR) and certain domain-specific datasets (Photo, Pubmed) while being less stable on small-scale datasets such as Texas and Wisconsin.

Table 8: Relations and Descriptions for Different Domains. "Relation type" of the knowledge graph is the corresponding dataset.

| Domain | Relation Type | Description |
|---|---|---|
| **Citation** | Citation | A citation in a network represents the act of one scientific paper referencing another, embodying the flow and accumulation of knowledge. It reflects the topical relevance between papers and researchers' acknowledgment of prior work, serving as a means to validate and support their own research while highlighting the collaborative nature of scientific inquiry. |
| | Paper classification | Paper classification in graph learning focuses on predicting the categories or topics of individual papers within a graph. |
| **WebKB** | Hyperlinks | A hyperlink represents a relationship or connection between nodes (e.g., entities such as documents, web pages, or words). It encapsulates semantic associations, enables information flow, and reflects the nature of interactions, which may vary in type, intensity, or direction. |
| | Webpage classification | Webpage classification involves predicting the categories or types of individual webpages in a graph, often determined by the content, purpose, or role of the webpage within a network of hyperlinks. |
| **Amazon** | Co-purchase | The "bought together" relationship between goods represents the co-purchasing behavior of customers, indicating a semantic association between products based on shared purchasing patterns. This relationship reflects the likelihood of two products being bought in conjunction, capturing their complementary nature or relevance in a shopping context. |
| | Product classification | Product classification is a task in graph learning that predicts the categories or labels of product nodes, focusing on their classification based on attributes like type, function, or consumer category in a co-purchasing or recommendation graph. |
| **Knowledge Graph** | (FB15K237) | [ '/location/country/form_of_government', '/tv/tv_program/regular_cast./tv/regular_tv_appearance/actor', '/media_common/netflix_genre/titles', '/award/award_winner/awards_won./award/award_honor/award_winner', '/soccer/football_team/current_roster./sports/sports_team_roster/position', '/soccer/football_team/current_roster./soccer/football_roster_position/position', '/film/actor/film./film/performance/film', ... , ] (Total 237 relations) |
| | (WN18RR) | [ '_hypernym', '_derivationally_related_form', '_instance_hypernym', '_also_see', '_member_meronym', '_synset_domain_topic_of', '_has_part', '_member_of_domain_usage', '_member_of_domain_region', '_verb_group', '_similar_to' ] (Total 11 relations) |

# I  BROADER IMPACTS

The proposed framework REEF has significant broader impacts for the development of graph foundation models (GFMs). By introducing relation tokens and adaptive hypernetworks, REEF offers a flexible and scalable approach to handle relational information across diverse datasets. This methodology not only improves the generalization and transferability of graph-based models but also provides insights into the potential of multi-dataset learning and graph data augmentation. The ability to adaptively generate parameters for aggregators, classifiers, and feature projectors across different domains paves the way for more robust and transferable GFM applications in fields such as knowledge graph construction, recommendation systems, and multi-modal learning. Additionally, the identification of scaling laws in REEF can guide future research on scaling GFMs for larger, more complex datasets, contributing to the broader field of AI-driven graph learning.

## J    THE USE OF LARGE LANGUAGE MODELS (LLMs)

In this paper, large language models were utilized exclusively for grammatical polishing and stylistic refinement, aimed at enhancing the clarity and readability of our presentation of results and conclusions.

