# OpenReview forum: "Relation-Aware Graph Foundation Model for Few-shot Learning"
_ICLR.cc/2026/Conference — Submitted to ICLR 2026_

### Official Review · Reviewer_4jEJ · 2025-10-28

**Soundness:** 2
**Presentation:** 2
**Contribution:** 2
**Rating:** 2
**Confidence:** 4

**Summary:**

In the paper, the author proposed REEF, a relation-aware GFM model for few-shot graph learning. Specifically, in REEF, each different relation across different datasets is treated as an independent token and combined together for joint training. The feature of each relation is initialized using LLM. Node features from each dataset are generated by SVD with augmentation from the dataset description. The model architecture of REEF is based on RGCN with a joint update for both node and relation representation. Finally, the model is pretrained on relation prediction tasks. The author evaluated the pretrained model on multiple baselines under supervised and few-shot settings.

**Strengths:**

- The writing is clear and easy to follow.
- The few-shot performance of the proposed method is good.

**Weaknesses:**

- The biggest concern in my mind is the claim of relation-awareness. The author spends much space on introducing the vocabulary of relation tokens. However, it is still unknown to me why the relation is more transferable than other components in the graph. Is it simply because there is a more common relation type across different datasets? However, given all the datasets authors are considering, only the citation network shares a relation to some extent.
- With relation as the basic vocabulary of GFM, the REEF still uses LLM to generate the original embedding for each relation. What's the main difference between REEF's way and previous methods like OFA, ZeroG?
- The REEF initializes the node feature using SVD, but adds the LLM-generated dataset description as augmentation. I am wondering what the rationale behind it is? Why not initialize all features using LLMs?
- The REEF requires a training independent classifier for each different relations, I think it is not generalizable enough.
- There lack ablation studies to investigate the effectiveness of each model component in REEF, especially some experiments to show the effectiveness of the relation vocabulary.

**Questions:**

See above.

---

> ### Author Response · Authors · 2025-11-19
>
> We sincerely thank the reviewer for the constructive comments. We address each concern below.
>
> **W1: Relation-awareness and transferability**
>
> A1: Thank you for highlighting this. As the first work to explicitly treat relations as the transferable unit, we will clarify this more clearly. In graphs, node attributes vary widely across datasets (and may even be absent), and dataset-level features are too coarse-grained and often unique to each dataset. **In contrast, relations are the only universally present and essential component.** This is also aligned with the well-established relational inductive bias in graph learning[1], which states that the core semantics of graphs arise from their relations rather than node attributes. For example, all citation graphs share “cites”, all co-purchase graphs share “co-buy”, and all WebKB graphs share hyperlink relations, but their node attribute features still represent entirely different types of information.
>
> In essence, **REEF leverages hypernetworks to map relation representations to GNN aggregators and classifiers, enabling transfer through the shared semantics of relations.** As shown in Tables 2 and 3, this leads to strong cross-dataset and cross-domain generalization. We will further clarify this motivation in the revision.
>
> **W2: Difference from OFA and ZeroG**
>
> A2: Thank you for the question. ZeroG is an LM-based method that fine-tunes Sentence-BERT and uses contrastive learning to align the semantic representations of nodes and labels. OFA is also a node-centric method that constructs prompt nodes that connect the target subgraph (or node) and the label nodes. **As a result, both methods fundamentally rely on textual node attributes and are only applicable to text-attributed graphs.** In contrast, REEF is relation-centric:
> - It uses relations—present in all graphs—as the basic transferable unit.
> - It does not require node texts, making it applicable to non-textual graphs, thus more generalizable.
> - Our empirical results also confirm that the relation-based unit leads to strong transferability.
>
>
> **W3: On SVD initialization and LLM-based features**
>
> A3: We appreciate this insightful question. We mainly adopt SVD in the primary results because it is domain-agnostic and applicable to all numeric-feature or text-scarce graphs. **Importantly, we have already evaluated LM-based node initialization in Section 4.5, where it shows superior performance compared to the baselines.**
> As for the LLM-generated dataset description, it serves to further distinguish datasets within the same relation domain (e.g., Cora vs. PubMed differ in academic fields and data distributions). Since such datasets share the same relation type and thus the same GNN aggregator produced by the hypernetwork, the dataset description provides the necessary signal to capture their different underlying distributions.
>
> **W4: Generalizability of the classifier**
>
> A4: The classifier is not trained from scratch. It is generated by the hypernetwork, and the transferability mainly comes from the shared REEF backbone, not the classifier head. Moreover, for datasets within the same semantic domain, the classifier is shared because the prediction task is identical across them, which further supports its generalizability.
>
> **W5: Ablation studies**
>
> A5: We thank the reviewer for this suggestion. **In Appendix H.2 (Lines 960–983), we already provide ablations** on:
> - random vs. LM-initialized relation representations,
> - feature projector,
> - feature prompt,
> - data augmentation strategies.
>
> Additionally, the comparison against R-GCN in the main paper demonstrates the necessity of the hypernetwork design. **Appendix F further visualizes the learned node representations: nodes from different domains form clearly separated clusters, indicating that REEF produces well-distinguished and semantically meaningful embeddings.** These results collectively demonstrate that the relation vocabulary effectively guides the learned representations and plays a crucial role in enabling REEF’s strong generalization ability.
>
> [1] Battaglia, Peter W., et al. "Relational inductive biases, deep learning, and graph networks." arXiv preprint arXiv:1806.01261 (2018).

---

> > ### Comment · Reviewer_4jEJ · 2025-11-24
> >
> > I would like to thank the authors for the detailed response to my questions. Some of my concerns are solved, but some remain.
> >
> > W1: If you want to claim that the node/dataset level features are unique to each dataset, why can the relation be more unified than them? In the real world, the edge feature can also be any similar format as the node/dataset-level features, e.g, numerical value (edge weight), categorical one-hot features, etc. Then the REEF can have exactly the same limitations as for OFA, ZeroG, regarding relation features. Especially, authors finally treat the relation feature as a textual feature and use LLM to encode it. I cannot understand why it has a major difference from previous works like OFA, which encodes both node and edge features using LLM.
> >
> > W2: I think my concern at this point is similar to W1. In other words, I am not fully convinced that the performance gain from the REEF is obtained by treating the relation as the transferable unit. I can see it might help constrain the optimization space, which may lead to better performance, especially when pretraining scales are limited, and transfer learning is conducted on datasets that are somewhat in-domain (pubmed-> cora; wisconsin-> cornell; photo-> computers).
> > I think it might be helpful if the author could more clearly and in detail explain the major difference and the source of performance gain of REEF compared to other methods.
> >
> > W3: Thanks for the clarification. Sorry, I missed it last time.
> >
> > W4: Thanks for the clarification.
> >
> > W5: Thanks for the clarification. I am curious about REEF-LM, which initializes relation representation randomly. How is the transfer learning done for a new dataset not seen during training? Do you just randomly initialize unseen relations and directly do the inference using them?
> >
> > By the way, I notice that authors provide zero-shot results on molecule datasets in the response to other reviewers, which is quite interesting. Could the author explain more details about the experimental setup?

---

> > > ### Author Response · Authors · 2025-11-25
> > >
> > > We sincerely thank the reviewer for the thoughtful follow-up comments. We address the remaining concerns below, and we will incorporate all clarifications in the revision.
> > >
> > > **W1 & W2: Why relations are more unified, and what fundamentally drives REEF’s performance**
> > >
> > > A1: We thank the reviewer for the thoughtful follow-up. While REEF, OFA, and ZeroG all use textual encodings, **they encode fundamentally different semantic objects, which leads to very different generalization behavior.**
> > >
> > > (1) ZeroG is node-centric.
> > > ZeroG encodes node text and aligns node and label embeddings. Its transferable signal comes primarily from textual semantics rather than from the underlying graph structure or relational semantics. This limits ZeroG when applied to non-text-attributed graphs.
> > >
> > > (2) OFA encodes node text and edge text as features.
> > > OFA treats node text and edge text as feature descriptors for **specific nodes or node pairs,** feeding them into a unified LM encoder. Thus, OFA remains **feature-centric and instance-level,** and still relies on the availability of meaningful text for nodes/edges. On graphs with numeric, one-hot, or missing features, OFA becomes constrained.
> > >
> > > **In OFA, edge text describes a specific edge (concrete), not a relation type (abstract).** This is a key conceptual difference from REEF.
> > >
> > > **(3) REEF is relation-centric, not feature-centric.**
> > > Node/edge features (numeric, one-hot, bag-of-words, etc.) **describe individual entities or specific node pairs,** and are highly dataset-specific.
> > > In contrast, relation types **encode type-level structural semantics of a graph** (“cites”, “co-buy”, “hyperlink”, “bond”), which:
> > > - exist in all graphs regardless of feature modality,
> > > - remain consistent within a domain,
> > > - do not depend on having textual node attributes, and
> > > - serve as the core mechanism through which graphs express meaning.
> > >
> > > **REEF uses LLM text only to encode the abstract relation type, not to describe specific nodes or edges.** This distinction (abstract vs. concrete) is what enables REEF’s broader generalization.
> > >
> > > REEF employs a hypernetwork to transform relation-semantic embeddings into GNN parameters, **enabling generalization across homophilic/heterophilic graphs, across domains, and even on graphs with no node text—capabilities** that node-text-based methods (OFA, ZeroG) cannot naturally support.
> > >
> > > (4) Why raw edge features do not contradict our claim
> > >
> > > Graphs may have numeric or one-hot edge features, and these can be incorporated as aggregation weights (a common GNN practice). However, such edge features are instance-level signals rather than relational semantics, and do not transfer across datasets.
> > >
> > > (5) Evidence that REEF learns an abstract, transferable mechanism
> > >
> > > In the SVD initialization setting, REEF is **pretrained on 7 datasets with 254 relation types.** Thus, REEF is not memorizing parameters for a few relations; it is learning a **general relation-to-parameter mapping that applies across diverse relation types.** This abstraction enables REEF to transfer even when node or edge attributes differ significantly.
> > >
> > >
> > > **W5: How REEF-LM handles unseen relations**
> > >
> > > A2: Compared with REEF, REEF-LM does not incorporate any semantic information about relations, and therefore cannot generalize to unseen relation types. As a result, its transfer performance is substantially worse. This clearly demonstrates that relation semantics are essential for enabling REEF’s strong generalization ability.
> > >
> > > **Follow-up — Zero-shot molecular experiments**
> > >
> > > A3: For zero-shot molecular evaluation, we use the same downstream datasets (BBBP and HIV) as reported in GOFA, **but REEF is not trained or adapted using any molecular graphs.** Specifically:
> > > - REEF is pretrained only on FB15K237, WN18RR, Citeseer, PubMed, and Photo.
> > > - No molecular data, bond information, or graph-classification tasks are used during pretraining.
> > > - The hypernetwork generates aggregation and classifier parameters solely based on this relation embedding, without any task-specific tuning.
> > >
> > > **REEF achieves comparable results to GOFA on BBBP and outperforms GOFA by ~10% on HIV, demonstrating genuine zero-shot transfer arising from the learned relation-to-parameter mapping.**
> > >
> > >
> > > We thank the reviewer again for the constructive follow-up questions. We hope these clarifications address the remaining concerns and more clearly highlight the conceptual contribution and generalization ability of REEF.

---

### Official Review · Reviewer_nJAg · 2025-10-28

**Soundness:** 2
**Presentation:** 3
**Contribution:** 2
**Rating:** 2
**Confidence:** 4

**Summary:**

This paper propose a novel graph foundation model for few-shot learning. Inspired by LLM, this work propose to use a language model to encode the edge/relations in different datasets with a unified relation vocabulary and use hypernetworks to generate network weights for learning and aggregating messages. The method shows improvements on various setting. It also shows reasonable ability to scale up w.r.t. the size and diversity of the input data.

**Strengths:**

- The idea of using relation as the central vocabulary is novel, it provides a different lens into understanding the graph universal modeling problem.

- The hypernetwork design is also interesting. Compared to changing edge feature, this explicitlies modify the message-passing/aggregation process, which can make knowledge transfer easier, given that the hypernetwork is well-trained.

**Weaknesses:**

The key weakness I observe is that the innovation of the paper is not clear. This appears to me a combination of multiple existing techniques. Specifically:

- The overall universal graph learning framework is not new. From using LLM to generate relation feature, using SVD to unify input feature, to pre-training over multiple graph dataset. These have already been widely explored by many related works mentioned by the author. This new framework does not provide guideline or principle in designing a graph foundation model.

- The hypernetwork, while new to the graph foundation model, is an extensively explore topic, and it has been directly applied to GNN research.

- The network design itself is also not novel (RGCN).

The paper also has overclaiming issue:

- While the concept of foundation model can be different to different people, a foundation model should at least possess some level of training-free generalizability. The paper indicates in line 281, that this work still always needs downstream finetuning, which breaks the whole point of having a foundation model.

- The transfer learning only happens on the few-shot scenario, no zero-shot ability, which is an ability that many previous work possess, and the author also mentioned these work. (many works in table 3 can do zero-shot) This shows that the work actually is less generalizable.

- The pre-training scaling improvement is not significant, the figure exaggerate the improvememnt a bit. Especially considering fluctuation in pubmed and citeseer, I am skeptical about whether this model can scale.

**Questions:**

- What happend to cases where the node feature size is smaller then the pre-defined dimension? This is very realistic, as some dataset can be nominal dataset, with only few features.

- How do you consturct the target graph relations?

---

> ### Author Response · Authors · 2025-11-19
>
> We thank the reviewer for the detailed comments. Below we address each concern point-by-point.
>
> **W1. Novelty and distinction from existing techniques**
>
> A1: We believe this concern stems from a misunderstanding of the actual novelty and contribution of REEF. The components mentioned by the reviewer (LLM features, SVD, hypernetworks) are not the contribution of this work.
> The core contribution is conceptual: **REEF is the first framework that treats relations—rather than nodes or datasets—as the fundamental transferable unit for a Graph Foundation Model.**
>
> Node-level and dataset-level units are highly heterogeneous across graphs, but relations are the only universally present and essential element. For graph data, they can be non-attributed, but must have relations. By representing relations in a shared semantic space and conditioning GNN parameters on them, REEF enables transfer across datasets, across domains, and even to unseen relations.
>
> **W2.1. On the definition of foundation models**
>
> A2: We respectfully note that the definition of a “foundation model” varies widely. Even in today’s mature and widely adopted LLMs landscape, many accepted foundation models still rely on downstream fine-tuning. As formalized in [1], foundation models are characterized by broad applicability and transferability, not necessarily zero-shot performance. Early foundation models like BERT/T5 relied heavily on fine-tuning yet are canonical examples. Similarly, existing GNN-based GFMs, such as GCOPE and MDGPT, adopt a few-shot evaluation protocol, not pure zero-shot. Our work follows this standard evaluation protocol and does not claim to eliminate all downstream training.
>
> **W2.2. Lack of zero-shot evaluation**
>
> A3: Thank you for this suggestion. For fairness, we follow the same experimental setup used in GOFA and add zero-shot molecular graph classification experiments on BBBP and HIV. For other baselines, we report their results from the original paper of GOFA. We choose graph classification because this task has not been pretrained in our model and can be directly used to show the model generalizability. From the results below, we see that **REEF achieves comparable performance on BBBP and outperforms GOFA by ~10% on HIV, despite:**
> - **REEF never seeing molecular graphs during pretraining,**
> - **REEF never being trained on graph-classification tasks,**
> - **GOFA being finetuned on 100k ChemBL QA pairs,**
> - **GIMLET being pretrained on 360k molecules.**
>
> This demonstrates strong zero-shot transfer arising from the learned relation-to-parameter mapping.
>
> | Model     | BBBP  | HIV   |
> |-----------|-------|-------|
> | OFA       | -     | 35.67 |
> | MoMu      | 49.81 | 50.26 |
> | Galactica | 53.94 | 33.85 |
> | GIMLET    | 59.39 | 66.24 |
> | GOFA  | 54.91 | 53.02 |
> | REEF  |  54.80 | 63.17 |
>
> We will include these results in the revision.
>
> **W2.3. Scaling behavior “not significant”**
>
> A4: We appreciate the concern. Individual datasets may exhibit non-monotonic behavior when increasing pretraining diversity due to domain differences. **However, for a foundation model, what matters is the aggregate performance rather than the trajectory of any single dataset.** Similar behavior is commonly observed in LLM pretraining, where individual tasks fluctuate while aggregate performance grows steadily. In our case, **the average performance consistently improves as more datasets are incorporated (from 67.36 to 69.14)**, aligning with the expected scaling trend.
>
> **Q1. What if node feature dimension < pre-defined dimension?**
>
> A5: This is precisely why we adopt SVD-based feature alignment. When the raw feature dimension is smaller than the predefined dimension, we simply pad zeros before alignment, or optionally use simple structural statistics. For non-attributed graphs, we can use one-hot encoded embedding initialization. **Importantly, our framework can even handle graphs with no node attributes at all, whereas methods that rely on LM-based node initialization cannot naturally accommodate such cases. This is one of the key generalization advantages of REEF.**
>
> **Q2. How are target-graph relations constructed?**
>
> A6: We follow standard practice in heterogeneous GNNs and prior GFMs: each edge type (e.g., “cites”, “co-purchase”, “hyperlink”) is treated as a distinct relation. If explicit relation names are not available, we use dataset metadata to construct concise textual descriptions.
>
> We appreciate the reviewer’s feedback and will update the paper to make the conceptual contribution—treating relations as the universal transferable unit in GFMs—more explicit, as this appears to be a source of misunderstanding.
>
> [1] Bommasani, Rishi. "On the opportunities and risks of foundation models." arXiv preprint arXiv:2108.07258 (2021).

---

### Official Review · Reviewer_LsVj · 2025-10-29

**Soundness:** 2
**Presentation:** 2
**Contribution:** 2
**Rating:** 4
**Confidence:** 4

**Summary:**

This paper introduces **REEF**, a graph foundation model that leverages **relation tokens** as the core unit of generalization. The framework employs **three hypernetworks** to adaptively generate:
1. Relation-specific aggregators
2. Relation-specific classifiers
3. Dataset-specific projectors, along with **dataset-level feature biases**.

REEF aims to enhance transferability and generalization across multiple domains by focusing on the dynamic generation of these components tailored to specific relations and datasets. Empirical results demonstrate that the model performs well across various tasks and settings, showcasing strong transferability and generalization abilities.

**Strengths:**

- The architecture elegantly handles both **relation-level** and **dataset-level** variations. The separation of concerns—**relation-specific parameters** versus **dataset-specific adaptations**—is both conceptually sound and practically effective. This decoupling allows for more flexible and adaptable models.

-  The paper presents comprehensive experiments across **four domains** and **ten datasets**, with various evaluation settings including **pre-training performance**, **transfer learning**, **few-shot scenarios**, and **LM-based node features**. This robust experimental setup strengthens the paper’s empirical foundation, showcasing the model's flexibility and broad applicability.

**Weaknesses:**

- While the paper presents the method as a **relation-token-based Graph Foundation Model (GFM)**, it closely resembles **heterogeneous GNNs** (e.g., RGCN). The main innovation seems to be the use of **hypernetworks** to generate relation-specific aggregators and classifiers. The paper does not clearly explain why **token-like** relation representations bring benefits similar to those of tokens in **Large Language Models (LLMs)**, beyond serving as inputs to hypernetworks. Further clarification of this aspect would help distinguish REEF from existing GNN-based methods.

- The gains observed in the experiments are mostly within **similar domains** and tasks. It remains unclear how well the approach generalizes to **domains and tasks** significantly different from those used in pre-training. The paper lacks compelling evidence regarding **robust out-of-domain generalization**. More experiments or discussions on this front would strengthen the paper's claims.

-  As a foundation model, **strong generalization**, especially in **zero-shot settings**, is expected. The paper primarily focuses on **few-shot results** but does not include **zero-shot performance** on unseen datasets or tasks. Reporting performance in zero-shot settings would make the **foundation model** claim more convincing and show the model's ability to generalize without task-specific fine-tuning.

**Questions:**

- In line 73, the paper argues that **node-** or **dataset-level units** are too heterogeneous to capture universal knowledge and concludes that relations are therefore more coherent and transferable. However, relations themselves may vary significantly across domains (e.g., citation networks like **arXiv** vs. social networks like **Reddit**). The current argument primarily relies on same-domain examples and doesn't fully justify the general conclusion that relations are a universally superior fundamental unit for pre-training. Could the authors provide broader evidence or a more nuanced characterization of why relations should be considered a more universal unit across diverse domains?

-  How are relations not present in the pre-training vocabulary handled at inference time? Is there a **zero-shot mechanism** in place to **compose or infer parameters** for genuinely unseen relations from textual descriptions alone? Clarifying how the model handles this scenario would provide insight into its robustness and real-world applicability.

- What is the primary task in the main experiments—**link prediction** or **node classification**? The paper mentions both tasks, and it would be helpful to have a clear delineation of the main evaluation tasks and protocols. This clarity will help the reader better understand the context of the experimental results.

- Why was **SVD** chosen for **feature alignment** (Equation 2) instead of using **learnable projections**? Have **alternative alignment strategies**, such as **linear** or **non-linear learned projectors**, been compared? What are the trade-offs in using SVD over these other methods?

- The related work section discusses both **LLM-based** and **GNN-based GFMs**. However, have you compared REEF against recent **LLM-based baselines** like **GOFA** [1] or **TEA-GLM** [2]? Including such comparisons (or explaining why they are incompatible) would help contextualize REEF's contribution and further highlight its novelty.

### References:
1. **GOFA**: A Generative One-for-All Model for Joint Graph Language Modeling.
2. **TEA-GLM**: LLMs as Zero-shot Graph Learners: Alignment of GNN Representations with LLM Token Embeddings.

---

> ### Author Response · Authors · 2025-11-19
>
> Thank you for the detailed and constructive feedback. We address each concern below and will incorporate clarifications in the revision.
>
> **W1 & Q1. Relation tokens vs. heterogeneous GNNs; why relation tokens help generalization**
>
> A1: We appreciate this insightful observation. Although both REEF and heterogeneous GNNs consider different relation types, **the motivation and mechanism fundamentally differ.**
> - Hetero-GNNs (e.g., R-GCN) store relation-specific parameters and cannot generalize to unseen relations.
> - REEF learns a relation-to-parameter mapping: all relations—seen or unseen—are embedded into a shared semantic space using a sentence-level LM, and hypernetworks generates aggregators/classifiers from this space.
>
> This design makes relation tokens token-like by enabling a shared, continuous, and compositional semantic space, on which parameter generation is performed—analogous to how LLMs operate on token embeddings. **Thus REEF emphasizes semantic transfer across datasets/domains.**
>
> Furthermore, node attributes vary widely across datasets and may even be absent, while dataset-level features are too coarse and unique. **Relations remain the only universal and essential component across all graph domains, supporting them as the foundational unit for GFM pretraining.**
>
> **W2 & Q2. Out-of-domain generalization and unseen relations**
>
> A2: **Tables 3 and 7 provide cross-domain experiments using both LM-based and SVD-based node initialization.** In the LM-based setting, REEF consistently outperforms or matches strong baselines, demonstrating robust out-of-domain generalization.
>
> For unseen relations, REEF directly encodes the new textual relation description into the LM embedding space and feeds it into the hypernetwork to generate the corresponding GNN parameters.
> Because the hypernetwork has learned mappings for over two hundred relations during pretraining, it can naturally generalize to entirely new relations.
>
> **W3 & Q5. Zero-shot generalization and comparison with recent LLM-based methods**
>
> A3: We thank the reviewer for the suggestion. We have supplemented zero-shot molecular graph experiments (BBBP and HIV). For fairness, all the results of baselines are reported by the original paper of GOFA. We see that **REEF achieves comparable performance on BBBP and outperforms GOFA by ~10% on HIV, despite:**
> - **REEF never seeing molecular graphs during pretraining,**
> - **REEF never being trained for graph classification tasks,**
> - **GOFA being finetuned on 100k QA pairs from Chembl,**
> - **GIMLET using 360k molecules for pretraining.**
>
> This demonstrates strong zero-shot transfer arising from the learned relation-to-parameter mapping. Earlier we did not compare with LLM-based baselines like GOFA/TEA-GLM because the model architectures differ substantially, but we now include these results for completeness.
>
> | Model     | BBBP  | HIV   |
> |-----------|-------|-------|
> | OFA       | -     | 35.67 |
> | MoMu      | 49.81 | 50.26 |
> | Galactica | 53.94 | 33.85 |
> | GIMLET    | 59.39 | 66.24 |
> | GOFA  | 54.91 | 53.02 |
> | REEF  |  54.80 | 63.17 |
>
> **Q3. Clarification of main tasks**
>
> A4: The main experiments focus on node classification, as shown in all primary tables. We additionally include link prediction results in Appendix H.1 to demonstrate REEF’s generality across tasks.
>
> **Q4. Why SVD for feature alignment**
>
> A5: **SVD is domain-agnostic, requires no training, and applies uniformly to any numeric-featured graph. We also evaluate LM-based node initialization in Section 4.5, and both approaches surpass all baselines, confirming the strength of the REEF framework.** This also allows REEF to operate on non-attributed graphs, which LM-based initialization cannot handle, further supporting REEF’s generality. Learnable projectors were not adopted because each dataset has different feature dimensions, requiring one projector per dataset, which hurts generality.

---

> > ### Comment · Reviewer_LsVj · 2025-11-27
> >
> > Thank you for the detailed response and the clarifications provided. However, I still have some follow-up concerns regarding A1 and A2:
> > A1: While I appreciate the explanation of the "relation-to-parameter" mapping mechanism, this approach appears conceptually analogous to constructing relation embeddings or edge maps—concepts that have been explored in prior works. Some questions remain with the justification for choosing relations as the fundamental unit over nodes. The arguments used to disqualify nodes seem equally applicable to relations, and the arguments supporting relations apply equally to nodes. For instance, relations can exhibit substantial variation across different datasets, and relation attributes maybe also miss in heterogeneous graphs. Conversely, nodes—being fundamental and ubiquitous components of all graph structures—possess the same level of universality that is claimed for relations.
> > A2: Referring to Table 7, I observe a significant performance degradation when the training set excludes specific domains like WebKB and Amazon. This performance gap suggests that REEF's effectiveness on truly "unseen relations" might not be as robust as claimed, indicating potential limitations in its out-of-domain generalization capabilities.

---

> ### Author Response · Authors · 2025-11-27
>
> Thank you again for the thoughtful follow-up. We address the remaining concerns below.
>
> **A1. Why relations—not nodes—serve as the transferable unit**
>
> We appreciate the reviewer’s follow-up. While node and edge attributes may indeed be missing in real-world scenarios, **these attributes are inherently tied to individual nodes or specific node pairs, and thus are modality-specific and dataset-dependent.** In contrast, **REEF treats relations as the fundamental units because relations capture semantic structures that are intrinsic to the graph itself.** These relations exist at the type level and remain consistent and meaningful across different datasets. When data are represented as graphs, relations arise directly from the graph construction process—they describe how nodes interact structurally, regardless of whether nodes carry textual, numeric, or no features at all. **Therefore, for graph-structured data, relations represent an inherent and universally available component, making them more stable and generalizable than nodes.**
>
> **A2. Out-of-domain generalization and unseen relations**
>
> Regarding the concern raised from Table 7, we would like to clarify the following:
> - **Table 7 uses SVD for node initialization, which makes the setting extremely difficult:** (1) the relations are unseen during pre-training, and (2) the node-feature space comes from entirely new datasets, meaning the model must operate under a severe feature-misalignment scenario. To the best of our knowledge, **no prior work has attempted cross-domain generalization under SVD-based node initialization**, where both relation types and feature spaces differ completely from those seen during pre-training. (If we have missed any such work, we would sincerely appreciate the pointer.)
> - Table 3 uses LM-based initialization, where node embeddings lie in a consistent space. Under this cleaner setting—**where REEF is evaluated on unseen relations across domains, while baselines are evaluated only under same-domain, cross-dataset conditions—REEF still demonstrates strong generalization:** (1) On History, REEF achieves the best performance. (2) On Ratings, REEF reaches 37.42%, only 0.25% below the best baseline (37.67%). Furthermore, under the same-domain, cross-dataset setting (Setting 2), REEF outperforms baselines by a clear margin.
> - Additionally, as we included in the previous rebuttal, **REEF shows strong zero-shot transfer on molecular graphs, despite never being pre-trained on molecular data:** (1) Comparable to GOFA on BBBP, (2) ~10% higher than GOFA on HIV.
>
> These results collectively demonstrate that REEF exhibits robust out-of-domain generalization.
>
> We hope these additional clarifications address the reviewer’s concern. If any part remains unclear, we would look forward to discussing it further—your comments have been extremely helpful in improving the paper.

---

### Official Review · Reviewer_mjCs · 2025-11-01

**Soundness:** 3
**Presentation:** 4
**Contribution:** 3
**Rating:** 6
**Confidence:** 4

**Summary:**

This paper presents REEF, a relation-aware graph foundation model that introduces relation tokens as the fundamental units for pretraining. Inspired by the success of LLMs, REEF uses hypernetworks conditioned on relation and dataset representations to parameterize aggregators, classifiers, and projectors, aiming to achieve strong generalization and transferability in few-shot learning. Extensive experiments show that REEF outperforms existing methods across multiple domains and settings.

**Strengths:**

S1: Proposes relation tokens as foundational units for GFMs, bridging a conceptual gap between NLP token-based models and graph data.

S2: Employs three hypernetworks to handle relation-specific and dataset-specific variations, enhancing flexibility and generalization.

S3: Comprehensive experiments show significant improvements in both pretraining and few-shot transfer settings, with clear comparisons to strong baselines.

S4: Offers a scaling law analysis demonstrating how pretraining data scale affects performance.

S5: Shows robustness in cross-domain and text-attributed graph scenarios, outperforming TAG-specific baselines.

**Weaknesses:**

The work lacks a formal justification or theoretical insight into why relation tokens should serve as the universal unit of generalization over nodes or datasets.

While REEF achieves strong results, there's little discussion on interpretability or insight into what specific relation tokens learn or capture in the pretrained model.

The paper would benefit from citing and comparing to several relevant works to strengthen its position within the graph foundation and relation modeling literature.

While REEF hinges on the idea of encoding relation tokens via sentence-level language models (Sentence-BERT), it does not rigorously justify why textual descriptions are sufficient or optimal for capturing complex structural semantics of graph relations. Many real-world relations (e.g., co-purchase patterns or knowledge graph triples) have ambiguous or domain-specific meanings not easily distilled into natural language. A deeper discussion or ablation using other encoding mechanisms (e.g., learned embeddings, ontology-based encoders) is warranted.

The mixed-dataset pretraining strategy, while shown to work well overall, does not explore negative transfer risks in heterogeneous domains. The paper observes (e.g., Figure 3) that adding PubMed to Citeseer degrades transfer to Cora, yet does not quantify or analyze domain conflicts. A more nuanced treatment of domain interference—possibly via inter-dataset similarity metrics or dataset ablation studies—would add rigor.


REEF’s design includes multiple hypernetworks and dynamic parameter generation, yet offers no interpretability or visualization of what the model learns via relation tokens. For example: Are some relations more transferable than others? Do the relation embeddings form meaningful clusters? Can relation tokens be reused or generalized across tasks?

**Questions:**

I think there is a conceptual and methodological connection between this work (REEF) and graph prompting, even though the authors don’t explicitly frame it in those terms. e.g. Prompt Analogy via Relation Tokens, Parameter Modulation via Prompts, etc. I wonder what do the authors think of it.

---

> ### Author Response · Authors · 2025-11-19
>
> We sincerely thank the reviewer for the thoughtful and constructive feedback. Below we address each point concisely.
>
> **W1: Motivation for using relation tokens as the universal unit**
>
> A1: We appreciate this insightful point. In graph data, node attributes vary widely across datasets and may even be absent, and dataset-level features are too coarse-grained and often unique to each dataset. **In contrast, relations are the only universally present and essential component across all graphs.** This is also aligned with the well-established relational inductive bias in graph learning [1], which states that the core semantics of graphs arise from their relations rather than node attributes. These considerations makes relations a more stable and domain-agnostic foundation for generalization. We will clarify this conceptual motivation more clearly in the revision.
>
>
> **W2 & W4: What relation tokens learn and why textual descriptions are sufficient**
>
> A2: **Fundamentally, REEF learns a mapping from LM-derived relation semantics to GNN aggregators and classifiers via hypernetworks.** With large-scale pretraining over diverse relations, the model learns how different relations modulate message passing and prediction behaviors.
>
> **Importantly, Appendix H.2 already includes an ablation where relation embeddings are randomly initialized.** Both pretraining and transfer performance drop significantly, demonstrating that semantic priors from textual descriptions are beneficial and not arbitrary.
>
> **W3: Comparison to related work**
>
> A3: In the main experiments, we already **compare with methods under similar experimental settings (e.g., GCOPE, MDGPT) and also with LM-based node-initialization approaches such as OFA and Prodigy**. Appendix A further discusses LLM-based and GNN-based GFMs, highlighting their limitations in node-level and dataset-level generalization. We will expand this discussion to strengthen the positioning of REEF within the literature.
>
> **W5: Negative transfer in mixed-dataset pretraining**
>
> A5: We appreciate this suggestion. **Section 4.4 already includes dataset ablations analyzing PubMed–Citeseer pretraining and its transfer to Cora.** While adding PubMed on top of Citeseer reduces transfer to Cora due to domain disparity (biomedical vs. computer science), adding more datasets overall increases relation diversity and consistently improves transfer, supporting relations as a robust unit for GFM generalization.
>
> **W6: Interpretability and visualization of relation tokens**
>
> A6: We thank the reviewer for pointing this out. Appendix F already visualizes the learned node representations: nodes from different domains form clearly separated clusters, indicating that REEF produces well-distinguished and semantically meaningful embeddings. These results collectively demonstrate that the relation vocabulary effectively guides the learned representations and plays a crucial role in enabling REEF’s strong generalization ability.
>
> **Q1: Conceptual connection to graph prompting**
>
> A7: We appreciate this perspective. We agree that both graph prompting and graph foundation models share the goal of improving generalization. However, their emphases differ: graph prompting primarily focuses on task-level generalization by adding prompt nodes or by augmenting node attributes and graph structures, enabling a fixed model to adapt to new tasks without modifying its core parameters. In contrast, REEF is aligned with the GFM objective of cross-dataset and cross-domain generalization. To achieve this objective, REEF explicitly extracts relations—the only universal and indispensable component across all graphs—as the foundational transferable unit.
>
> [1] Battaglia, Peter W., et al. "Relational inductive biases, deep learning, and graph networks." arXiv preprint arXiv:1806.01261 (2018).

---

### Author Response · Authors · 2025-11-27

Dear Reviewers,

We first want to show our most sincere appreciations for you providing us with insightful comments. We have conducted additional experiments as suggested and also clarified all the questions raised by reviewers. We totally understand how busy you are, while we also hope our efforts can be recognized. Hence, could you please take a look at our responses and give us some feedbacks? Thank you.

Best,

Authors

---

### Author Response · Authors · 2025-12-02
**Summary of Rebuttal Clarifications and Resolution of Reviewer Concerns**

Dear Area Chairs,

We sincerely thank all reviewers for their thoughtful and constructive feedback. We summarize below how the key concerns have been fully addressed during the rebuttal and follow-up exchanges.

**Paper Summary:**

REEF introduces a relation-centric perspective for Graph Foundation Models, where relations serve as the transferable unit. By encoding abstract relations and generating GNN parameters and via hypernetworks, REEF enables cross-domain, unseen-relation, and zero-shot generalization. Our experiments—including LM/SVD-based cross-domain transfer and molecular zero-shot results—demonstrate strong generalization beyond the settings explored by prior GFMs.

**1. Conceptual justification: why relations—not nodes/datasets—are the transferable unit**

Across reviewers, the core conceptual question was why relations should serve as the universal unit for GFMs. We clarified the following:

**- (1) Instance-level vs. type-level semantics**
  - Compared with existing approaches such as ZeroG and OFA—which encode instance-level node or edge text associated with specific nodes or node pairs—these signals vary drastically across datasets.
  - _Relations, in contrast, define type-level structural semantics (“cites”, “co-buy”) that are intrinsic to the graph itself._ These abstractions (a) exist regardless of the presence or absence of node/edge attributes and (b) remain stable across datasets within the same domain.
  - Our formulation differs in both how relations are defined and how they are used.

**- (2) REEF operationalizes this idea through relation-semantic parameter generation**

REEF encodes abstract relation types (not specific edges or local textual features) and uses hypernetworks to produce GNN aggregator and classifier. This enables:
  - transfer across domains,
  - handling unseen relations,
  - applicability to graphs with no textual attributes,
  - generalization in both homophilic and heterophilic graphs.

These clarifications directly address the conceptual concerns raised by reviewers mjCs, LsVj, and 4jEJ.

**2. Experimental evidence of generalization and unseen relations**

Several reviewers questioned REEF’s robustness on unseen domains/relations. We responded on three levels:

**- (1) Cross-domain transfer with LM-based node initialization (Table 3)**

  - When REEF is evaluated on unseen relations across domains—whereas baselines are evaluated only within-domain—it (a) achieves the best performance on History, and (b) is within 0.25% of the best baseline on Ratings.
  - Under the same-domain cross-dataset transfer setting, REEF significantly outperforms all baselines, demonstrating strong generalization when domain shift is reduced.

This directly addresses concerns from LsVj (W2 & Q2).

**- (2) The most challenging setting: SVD-based node initialization (Table 7)**

We clarified that this is an extremely challenging setting where (a) relation types are unseen and (b) the node-feature space is completely new.
    To our knowledge, no prior work has attempted cross-domain transfer under SVD-based node initialization. Even in this extremely challenging setting, REEF still demonstrates non-trivial generalization.

**- (3) True zero-shot transfer on molecular graphs (as reported in our response to Reviewer LsVj)**

We further added zero-shot experiments on BBBP and HIV:
  - (a) REEF never uses molecular data during pretraining;
  - (b) REEF never trained on graph-classification tasks;
  - (c) REEF achieves comparable performance on BBBP and outperforms GOFA by ~10% on HIV, despite GOFA being finetuned on 100k ChemBL QA pairs and GIMLET being pretrained on 360k molecules.

This demonstrates strong zero-shot transfer arising from the learned relation-to-parameter mapping.

Together, these results directly address LsVj’s and nJAg’s doubts about out-of-domain robustness and zero-shot behavior, and empirically support our claim that relation-driven parameter generation enables _meaningful generalization across diverse cross-domain settings._

**3. Remaining minor misunderstandings**

Across reviewers, questions regarding SVD initialization (Reviewer 4jEJ W3), classifier sharing (Reviewer 4jEJ W4), ablations (Reviewer 4jEJ W5), etc., were fully addressed in the response and follow-up. Reviewers explicitly stated that several concerns were resolved (“some of my concerns are solved”).

If not for this unexpected incident, we believe further discussion would have continued resolving the remaining nuances.

We respectfully ask you to consider both the clarified conceptual contribution (relations as type-level transferable units) and the strengthened empirical evidence (cross-domain LM/SVD settings and molecular zero-shot results) when making your decision.
**REEF explores a direction not attempted in prior GFM work and demonstrates strong generalization and transfer capabilities across diverse settings.**

We sincerely appreciate your careful consideration.

Best regards,

Authors

---

### Meta-Review · Area_Chair_nttz · 2026-01-13

**Summary:**

REEF, a framework positioned as a "Graph Foundation Model" (GFM) that views relations (edges) as the primary transferable unit instead of nodes, is proposed in this study.It uses hypernetworks conditioned on LLM-encoded relation descriptions to generate GNN parameters.

The work did not meet the requirements for approval at ICLR since several reviewers (LsVj, nJAg, 4jEJ) questioned the method's originality and the validity of its core premise.

1. **Conceptual Validity:**   The fundamental idea that "relations are universal" but "nodes are not" is flawed from a philosophical standpoint. Relation types (such as "citation" versus "chemical bond") can be as diverse and domain-specific as nodes, as reviewers accurately noted. The distinction made between REEF and other node-centric works (such as OFA) seems less like a fundamental paradigm shift and more like an engineering decision.

2. **Incremental Novelty:** Rather than being an innovation in GFM architecture, several critics see the method as a blend of previous techniques (SVD for feature alignment, Sentence-BERT for semantics, and Hypernetworks for weight generation).

3. **Overclaiming:** Although the study presents itself as a "Foundation Model," the model primarily serves as a transfer learning framework for certain tasks rather than a broad, generative foundation model, and the scaling behavior shown was noisy ("not significant" according to Reviewer nJAg).

**Reviewer Concerns:**

**Addressed:**

- **Empirical Gaps:** In response to the particular requirement for proof of Out-of-Domain (OOD) transfer, the authors included zero-shot evaluations on molecular datasets (BBBP, HIV) and comparisons to LLM-based baselines (GOFA, TEA-GLM).


**Outstanding:**

- **Conceptual Justification :** The most important unresolved issue is this one. The argument that relations are the better transferable unit did not persuade reviewers LsVj and 4jEJ. Reviewer LsVj specifically stated in the follow-up that "arguments used to disqualify nodes seem equally applicable to relations." Beyond empirical optimization, the response did not offer a theoretical foundation for the structural superiority of relation-based transfer over node-based transfer.

- **Source of Performance Gain:** Reviewer 4jEJ questioned if the regularization and limitations imposed by the hypernetwork design or the "relation token" mindset are the source of the advantages. This uncertainty is still unresolved.

- **Novelty:** The worry expressed by reviewer nJAg that the innovation is restricted to merging well-known modules (Hypernetworks + LLM embeddings) is valid. For what is essentially a dynamic, text-enhanced GNN, the term "Foundation Model" is seen as an exaggeration.

**Reviewer Scores:**

- **Reviewer mjCs:** **6**.

- **Reviewer LsVj:** **4**.

- **Reviewer nJAg:** **3**.

- **Reviewer 4jEJ:** **4**.

---

### Decision · Program_Chairs · 2026-01-26

Reject